# Impact of Long-Term Cold Storage on the Physicochemical Properties, Volatile Composition, and Sensory Attributes of Dried Jujube (*Zizyphus jujuba* Mill.)

**DOI:** 10.3390/foods14010050

**Published:** 2024-12-27

**Authors:** Shaoxiang Pan, Xuemei Liu, Xiaodong Zheng, Jihong Wu, Mengnan Tan, Ning Cao, Xinyu Zhao, Maoyu Wu, Yu Han, Xinhuan Yan, Ye Song

**Affiliations:** 1Jinan Fruit Research Institute, All China Federation of Supply & Marketing Co-Operatives, Jinan 250014, China; panshaoxiang@126.com (S.P.);; 2College of Food Science and Nutritional Engineering, China Agricultural University, Beijing 100083, China; 3Shandong Institute of Metrology, Jinan 250102, China

**Keywords:** grey jujube, storage, quality changes, descriptive sensory analysis, physicochemical-sensory correlation

## Abstract

Grey jujube (*Zizyphus jujuba* Mill. cv. Huizao), a prominent cultivar from Xinjiang, China, is well known for its high nutritional value and medicinal benefits. This study investigates the effects of long-term storage on the quality attributes of grey jujube, focusing on color, texture, physicochemical properties, bioactive ingredients, amino acid profiles, sensory characteristics, and volatile compounds. Over a three-year storage period, significant changes were observed, including a decline in lightness and redness of the peel, accompanied by textural modifications such as increased hardness and chewiness, primarily attributed to moisture loss. Physicochemical analyses revealed significant reductions in moisture content, sugars (particularly reducing sugars), and bioactive compounds such as vitamin C, total flavonoids, and cyclic adenosine monophosphate (cAMP). In contrast, total acidity showed a significant increase over time. Sensory evaluation indicated a substantial loss of fresh aroma and flavor, with the development of off-flavors over time. Additionally, a comprehensive analysis of volatile compounds highlighted significant transformations in aroma profiles, with notable reductions in desirable esters and increases in acetic acid concentrations. This study investigates the quality changes of grey jujubes during storage from sensory and physicochemical perspectives, aiming to provide a novel approach for differentiating between newly harvested and aged grey jujubes. Furthermore, these findings provide theoretical support for improving and optimizing storage conditions. Future research should focus on elucidating the underlying mechanisms of these changes, identifying key markers for quality control during grey jujube storage, and providing a scientific basis for distinguishing between newly harvested and aged grey jujubes, while improving storage quality.

## 1. Introduction

The jujube (*Zizyphus jujuba* Mill.), commonly referred to as the ash jujube, is a member of the Rhamnaceae family. Its cultivation in Xinjiang dates back to the early 1970s, when it was introduced from Xinzheng, Henan Province. Since then, it has become a dominant variety in Xinjiang’s fast-growing grey jujube industry [1]. Xinjiang has become a major hub for grey jujube production, particularly in regions including Aral, Kashgar, Ruoqiang, and Aksu. The superior quality of grey jujube produced in Xinjiang, attributable to the region’s unique light and thermal resources, has led to its economic and agricultural significance, surpassing that of the original cultivation area [2]. Grey jujube is valued as both a fresh and dried fruit, known for its compact texture, low moisture content, and high sugar concentration. Additionally, it possesses a distinct flavor profile and is abundant in essential trace elements and bioactive compounds, which contribute to its pharmacological effects, including sedative properties, hematopoietic support, antioxidant activity, and antineoplastic potential [1,3]. The fruit’s unique characteristics, including its high sugar content, rich flavor, have made it popular among consumers. With increasing consumer demand for higher fruit quality driven by rising living standards, growing interest has emerged in the nutritional and aromatic properties of fruits, key indicators of overall quality [4,5,6]. As living standards improve, there is an increasing demand for higher fruit quality, with particular attention from researchers on the nutritional and aroma components, which are key indicators of fruit quality.

Post-harvest, the majority of grey jujube production in Xinjiang is processed into dried fruit due to challenges in fresh storage and transportation [7,8]. Dried grey jujube is prized for its thick flesh, sweet and sour taste, and unique flavor, making it a popular market product [9,10]. As the cultivation area and production volume of grey jujube in Xinjiang continue to expand, market pressures have intensified, leading to unsold inventory during peak harvest seasons. To mitigate these challenges, more enterprises have adopted cold storage during the harvest period, enabling off-season sales or utilizing the fruit for further processing. Currently, grey jujubes sold on the market may be stored for up to three years. Grey jujubes that have been stored for long periods are visually similar to original jujubes, making it difficult to distinguish their storage duration by appearance alone, which poses significant challenges to the development of the industry. Despite its commercial success, storage remains a significant challenge. Over time, the nutritional components of grey jujube degrade, leading to diminished sensory and nutritional quality, which in turn results in substantial economic losses [11,12,13]. While much of the research to date has focused on optimizing drying techniques and evaluating different grey jujube varieties, few studies have focused on the effects of long-term storage on the fruit’s nutritional and aromatic profiles.

Given the economic and nutritional importance of grey jujube, it is crucial to understand the mechanisms of quality deterioration during storage. This study aims to investigate the dynamic changes in key nutritional components and volatile compounds of grey jujube over an extended storage period. The findings will offer valuable insights into optimizing storage conditions to preserve the fruit’s quality and prolong its shelf life, thereby providing a theoretical foundation for the improvement of post-harvest handling practices in the grey jujube industry.

## 2. Materials and Methods

### 2.1. Materials

The grey jujube samples used in this study, dried jujube (*Zizyphus jujuba* Mill.), were purchased from Xinjiang Jujube Industry Co., Ltd. (Hetian, China), in 2019, with 13.2% water content and 84.95% total sugar content. All fruits were selected after natural on-tree drying, featuring by a uniform red skin color. To ensure consistency across the samples, the grey jujubes were dried under controlled conditions and manually sorted for uniform size and color prior to experimentation. The samples were stored in a cold storage facility at 4 °C for three years, maintaining stable conditions throughout the study. During the storage period, samples were taken at regular intervals, yielding seven samples: the pre-storage control sample (Y0), samples stored for six months (Y0.5), one year (Y1), one and a half years (Y1.5), two years (Y2), two and a half years (Y2.5), and three years (Y3). Each sample weighed approximately 3 kg. The cold-stored samples were first used to select 40 fruits for measuring color and texture. Then, 500 g of the sample was taken, chopped, and thoroughly mixed. Next, a subsample was taken using the quartering method, frozen with liquid nitrogen, and ground into a powder using a grinder. The powder was stored in a polyethylene plastic bottle at temperatures below −20 °C for subsequent antioxidant measurements. The remaining samples were chopped, mixed thoroughly, and used for further physicochemical analyses.

### 2.2. Chemicals

A homologous series of n-alkanes (C8–C20) was utilized for the calculation of retention indices (RI) in the gas chromatography analyses. The experimental procedures, including RI calculations, were conducted following the manufacturer’s guidelines (Sigma Chemical Co., St. Louis, MO, USA). 2-Octanol, used as an internal standard at a concentration of 20 mg/kg, was obtained from J&K Chemical Ltd. (Shanghai, China). Concentrated hydrochloric acid, sodium hydroxide, methyl red indicator, Fehling’s reagent, potassium sodium tartrate, glacial acetic acid, zinc acetate, potassium ferrocyanide, sodium tungstate, sodium molybdate, sodium carbonate, copper sulfate, potassium sulfate, boric acid, concentrated sulfuric acid, bromocresol green indicator, methylene blue indicator, 95% ethanol, phenolphthalein indicator, metaphosphoric acid, sodium bicarbonate, folin–ciocalteu reagent, sodium nitrite, aluminum nitrate, oxalic acid, sulfosalicylic acid and 2,6-dichlorindole, all of analytical grade, were obtained from Sinopharm Chemical Reagent Co. (Shanghai, China). Methanol and acetonitrile (HPLC grade, purchased from Sigma-Aldrich, St. Louis, MO, USA). Adenosine cyclic monophosphate, rutin, gallic acid, vitamin C, fructose, glucose, sucrose, xylose, threonine, serine, glutamate, glycine, alanine, cystine, valine, methionine, isoleucine, leucine, tyrosine, phenylalanine, lysine, histidine, arginine, and proline were purchased from Sigma-Aldrich (St. Louis, MO, USA).

### 2.3. Color Measurement

Grey jujube color was measured using a HunterLab ColorFlex spectrophotometer (HunterLab, Reston, VA, USA). The instrument was calibrated before each use with standard black and white tiles (L* = 92.23, a* = −1.29, b* = 1.19). Color parameters including lightness (L*), the red-green axis (a*), and the blue-yellow axis (b*) were recorded [14].

### 2.4. Texture Testing Method

Whole grey jujube texture was measured using a Texture Analyzer (TA-XT Plus; Stable Micro Systems Ltd., Godalming, UK) equipped with a P/20 probe [15]. The test parameters were set as follows: pre-test speed = 3 mm/s, test speed = 3 mm/s, post-test speed = 3 mm/s, compression depth = 30%, compression time = 5 s, and trigger force = 10 g. whole grey jujube texture, including hardness, springiness, cohesiveness, chewiness, and resilience, were evaluated. Hardness was defined as the peak force during the second compression of each sample. Individual measurements were performed on each grey jujube sample group.

### 2.5. Chemical Composition Analysis

Moisture content, Total sugar content and reducing sugar content were determined as described by Xu [1]. Titratable acidity (TA) was determined from the collected filtrate through neutralization titration using 0.1 mol/L sodium hydroxide (NaOH) until a pH of 8.1 was reached. The results were expressed in grams of citric acid per liter (g/L).

### 2.6. Total Phenolic Content

The total phenolic content was determined using the Folin–Ciocalteu method [16]. A total of 5 g of the sample was placed in a 100 mL volumetric flask and diluted with ultrapure water to 80 mL. The flask was immersed in boiling water for 30 min. After cooling to room temperature, the solution was diluted to the final volume. The extracts were then centrifuged (Sorvall ST8, Thermo Fisher Scientific, Waltham, MA, USA) at 4690 g for 4 min. A 1 mL aliquot of the supernatant was mixed with 5 mL of deionized water, 3 mL of 7.5% sodium carbonate (Na_2_CO_3_) solution, and 1 mL of Folin–Ciocalteu reagent. The mixture was thoroughly vortexed and incubated in the dark at room temperature for 2 h. Absorbance was measured at 765 nm, and the results were expressed as milligrams of gallic acid equivalents per 1000 g of fresh weight (mg GAE/1000 g DW). The concentration range for the standard curve of gallic acid was 0.5–5.0 mg/L.

### 2.7. Total Flavonoid Content

The total flavonoid content (TF) was quantified using a spectrophotometric method according to a standard laboratory procedure [17]. A 5 g sample was transferred to a 100 mL beaker, and the pH of the solution was adjusted to 13.0 by titration with sodium hydroxide (0.1 mol/L NaOH). The mixture was shaken and allowed to sit for 30 min before being titrated to a pH of 6.0 using NaOH. The pH-adjusted supernatant was transferred to a 100 mL volumetric flask and diluted to the mark with ultrapure water. The solution was then filtered through filter paper. A 100 µL aliquot of the filtrate was mixed with 200 µL of 5% sodium nitrite (NaNO_2_) solution. After 6 min, 200 µL of 10% aluminum nitrate (Al(NO_3_)_3_) solution was added, and the mixture was left to react for another 6 min. Following this, 2 mL of 4% NaOH solution was added, and the final volume was adjusted to 5 mL with distilled water. The solution was thoroughly mixed and allowed to stand for 15 min before the absorbance was measured at 510 nm using a spectrophotometer. The total flavonoid content was expressed as milligrams of rutin equivalents per 100 g of fresh weight (mg RE/100 g DW).

### 2.8. Vitamin C Analysis

Vitamin C content was determined using the visual titration method based on the reduction of 2,6-dichlorophenol-indophenol dye [18]. A 100 g portion of the edible part of the sample was weighed and placed into a tissue grinder, followed by the addition of 100 mL of 2% oxalic acid extraction solution. The sample was quickly ground into a homogeneous slurry. A portion of 10 g of the slurry was transferred into a 100 mL volumetric flask, diluted to the mark with extraction solution, and mixed thoroughly. The solution was filtered, and 10 mL of the filtrate was transferred into a 50 mL conical flask. Titration was performed using a standardized 2,6-dichlorophenol-indophenol solution until the solution turned pink and maintained the color for 15 s without fading. All determinations were done in triplicate, and results were expressed as milligrams per 100 g DW.
(1)$$X=\frac{(V-V_{0})\times T\times A}{m}\times 100$$

In the formula, *X* represents the content of Vitamin C in mg/100 g, *V* represents the volume of dye solution consumed during the titration of the sample solution in mL, *V*_0_ represents the volume of dye solution consumed during the titration of the blank in mL, *T* represents the titration value of 2,6-dichlorindole dye (0.0952 mg/mL), *A* represents dilution factor, *m* represents sample weight in g, and 100 represents unit conversion factor.

### 2.9. Analysis of Amino Acids

Amino acids were extracted from a 2 g grey jujube sample using 60 mL of 0.266 mol/L sulfosalicylic acid solution at room temperature over a 12-h period. The resulting suspension was centrifuged at 2931× *g* for 10 min at 0 °C, then filtered through a 0.22 μm syringe filter. Amino acid content was quantified using the ninhydrin method on an automatic amino acid analyzer (A300, Membrapure, Berlin, Germany). All determinations were done in triplicate, and results were expressed as grams per 100 g DW.
(2)$$X_{i}=\frac{c_{i}\times F\times V\times M}{m\times 10^{9}}\times 100$$

In the formula, *X_i_* represents the concentrations of amino acids in g/100 g, *c_i_* represents the content of amino acid *i* in the sample solution in nmol/mL, *F* represents dilution factor, *V* represents final volume in mL, *M* represents molar mass of amino acid *i* in g/mol, *m* represents sample weight in g, 10^9^ and 100 represents the unit conversion factor.

### 2.10. Cyclic Adenosine Monophosphate (cAMP) Analysis

A 5 g sample was placed in a 50 mL volumetric flask and diluted to 40 mL with ultrapure water. The sample was soaked for 20 min, then subjected to ultrasonic treatment for 20 min at 60 °C. After cooling to room temperature, the mixture was diluted to the final volume. The extracts were centrifuged (Sorvall ST8, Thermo Fisher Scientific, Waltham, MA, USA) at 1759 g for 10 min, and the supernatant was subsequently filtered through a 0.45 µm membrane filter. cAMP analysis was performed using an HPLC system (U3000, Thermo Fisher Scientific, Waltham, MA, USA) equipped with a C18 column (Acclaim™ 120, C18, 250 mm × 4.6 mm, 5 µm) and a UV-Vis detector (TU-1810, Puxi, Shanghai, China). The injection volume was 20 µL, and the flow rate was maintained at 1 mL/min. A binary mobile phase system was employed, consisting of methanol (A) and 50 mM KH₂PO₄ (B) at a ratio of 1:9 (A). Detection was carried out at a wavelength of 258 nm, with the column temperature set at 30 °C. All experiments were performed in duplicate, and each sample analysis was conducted in triplicate, with the mean values reported.

### 2.11. Determination of Xylose, Fructose, Glucose and Sucrose

The sugar profile was quantified following the method of Carbonell-Barrachina et al. [19], with slight modifications. Samples were extracted with water and analyzed using high-performance liquid chromatography (HPLC) (U3000, Thermo Fisher Scientific, Waltham, MA, USA). The mobile phase consisted of 80% acetonitrile at a flow rate of 0.8 mL/min. Individual sugars were separated on a Hypersil GOLD Amino column (250 mm × 4.6 mm, 5 μm) and detected using a refractive index (RI) detector. Standards for glucose, fructose, sucrose, and xylose were obtained from Sigma (Poole, UK), and calibration curves were constructed for quantification, demonstrating high linearity (R^2^ = 0.999). Sugar concentrations were expressed as g per 100 mL of fresh weight (FW), and each analysis was performed in triplicate.

### 2.12. Gas Chromatography–Mass Spectrometry (GC–MS) Analysis

For each sample, 5 g of grey jujube puree was added to a 20-mL headspace bottle (Supelco, Bellefonte, PA, USA) supplemented with 20 μL 2-octanol of (32.88 μg/mL in methanol; Sigma-Aldrich, St Louis, MO, USA) as an internal standard. The sample vial was closely capped with a PTFE-silicon stopper and equilibrated at 40 °C for 10 min. Then, a DVB/CAR/PDMS fiber was inserted into the headspace with continuous heating and agitation (250 rpm) for 30 min. The SPME extract of the Y0, Y1, Y2 and Y3 samples were injected into the port of a Trace1300-ISQ (Thermo Fisher Scientific, Waltham, MA, USA) equipped with a DB-WAX capillary column (30 mm × 0.25 mm × 0.25 μm) and desorbed at 250 °C for 5 min.

GC–MS conditions were set in accordance with the procedure described by Yan et al. [20], with modifications. The injection port was operated in splitless mode and 99.999% pure helium was used for vial pressurization and as the carrier gas. The flow rate was 1 mL/min. The initial oven temperature was 40 °C (3 min), which was ramped up at 5 °C/min to 150 °C. Then it was ramped up at 3 °C/min to 240 °C and held there for 5 min. The mass detector was operated in the electronic impact (EI) mode at 70 eV and the source temperature was set at 230 °C. The mass spectra were scanned in the *m*/*z* 35–500 amu range.

The compounds were identified by matching retention times of retention indices (RIs) and mass spectra in the NIST 10 Database. The RIs of unknown compounds were determined by alkanes (C5-C30) (Sigma-Aldrich, St. Louis, MO, USA). Only compounds with similarities exceeding 85 (the maximum similarity is 100) are considered for identification. The quantitative identification of compounds of grey jujubes was determined by internal standard method (2-octanol, 20 mg/kg).

### 2.13. Gas Chromatography–Olfactometry(GC–O) and Aroma Frequency Detection Analysis

The odor-active compounds were characterized by a sniffing port (ODO II; SGE, Melbourne, Australia) coupled to a GC–MS (Trace1300-ISQ, Thermos) that is widely used in flavor analysis [21,22,23,24]. At the exit of the capillary column, the effluents were split 1:1 (by volume) into a sniffing port and an MS detector. The GC–MS conditions were the same as those described above. Aroma extracts were orthogonally evaluated twice by four trained sensory panelists. Retention time, odor characteristics, and aroma intensity (AI) were recorded by each trained sensory panelist.

### 2.14. Odor Activity Values (OAV)

The OAV of an aroma compound is equal to its concentration divided by its threshold concentration in water, which can be obtained from published values [25]. Compounds with OAVs ≥ 1 are considered potential contributors to the sample aroma profile [26,27].

### 2.15. Sensory Evaluation of Grey Jujubes

The sensory characteristics of the grey jujube samples were evaluated by a trained panel of 10 members (five males and five females) with extensive experience in food sensory evaluation, all recruited from the fruit and vegetable processing team. Organoleptic descriptors were quantified using six sensory attributes: “Jujube fragrance”, “Rancid”, “Sour”, “Sweet”, “Bitter”, and “Aftertaste” to assess both aroma defects and positive qualities. Each grey jujube sample was assigned a unique three-digit code and presented in random order to minimize bias. The evaluation was conducted at room temperature, with individual assessments recorded for each panelist.

The assessors used a five-point scale (0 = not detectable, 5 = very intense) with 1-unit increments to evaluate each sensory attribute. The assessors used a five-point scale (0 = not detectable, 5 = very intense) with 1-unit increments to evaluate each sensory attribute. A 3-min break was provided between the evaluation of each sample. This scale, favored for its simplicity and clarity, effectively captures a wide range of sensory intensities. Additionally, the scale includes enough discrete points to highlight subtle differences in intensity between samples. The results from three independent QDA tests for each odor descriptor were averaged and presented in a spider web chart.

### 2.16. Statistical Procedures

The significant differences in physicochemical components among the grey jujubes were determined by one-way analysis of variance (ANOVA) using SPSS (Version 25.0, IBM, Armonk, NY, USA). Duncan’s multiple test was applied to verify significant differences among the aroma compounds of three kinds of grey jujubes at a level of *p* < 0.05. The contents of different components were presented as the mean ± SD (standard deviation) of triple measurements. Cluster analysis was accomplished using Rstudio (R-Tools Technology, Inc., Richmond Hill, ON, Canada).

## 3. Results and Discussion

### 3.1. Physicochemical Properties

#### 3.1.1. Color Analysis

Color changes in the peel and flesh of grey jujubes during storage were evaluated and are presented in Figure 1. Compared to the initial storage sample (Y0), significant decreases in L* (lightness), a* (redness), and b* (yellowness) values of the peel were observed (*p* < 0.05) after two years of storage (Y2), indicating a progressive darkening and fading of red hues. Specifically, the peel L* value decreased from 31.29 ± 2.26 (Y0) to 26.99 ± 3.10 (Y2) and 27.11 ± 3.31 (Y3), while the a* value dropped from 24.20 ± 1.86 (Y0) to 17.38 ± 1.54 (Y3), and the b* value declined from 16.49 ± 1.84 (Y0) to 9.10 ± 1.24 (Y3). These changes suggest that the peel underwent noticeable darkening and loss of red pigmentation over time.

For the grey jujube flesh, a gradual decrease in L* was also observed, with values dropping from 69.34 ± 5.24 (Y0) to 61.24 ± 4.46 (Y2), reflecting reduced lightness. Conversely, the a* and b* values of the flesh increased over time, with the a* value rising from 8.42 ± 1.74 (Y0) to 11.76 ± 1.03 (Y3), and the b* value increasing from 31.55 ± 2.15 (Y0) to 35.83 ± 1.37 (Y3). These changes suggest a deepening of the flesh color, transitioning towards more intense red and yellow hues as storage progressed.

The observed color changes in both the peel and flesh of grey jujubes during storage may be attributed to several factors. Firstly, grey jujube contains oxidase enzymes, such as polyphenol oxidase (PPO) and catalase (CAT), which can be activated during storage. These enzymes catalyze the oxidation of phenolic compounds, leading to changes in pigment composition and a subsequent darkening of the fruit. For instance, naturally occurring flavonoids and anthocyanins in jujube may undergo oxidation, contributing to the color alterations observed. This enzymatic oxidation aligns with previous studies attributing color changes to the degradation of phenolic compounds. Additionally, the respiratory activity of jujubes during storage consumes sugars and generates carbon dioxide and ethylene. These metabolic processes can further induce chemical changes within the fruit, influencing the color stability of both the peel and flesh. Consequently, this oxidative degradation, combined with metabolic activity, likely contributes to the progressive darkening of the peel and the intensification of the flesh coloration, highlighting the impact of long-term storage on the appearance and quality of grey jujubes [28,29].

#### 3.1.2. Textural Analysis

The texture of grey jujube fruit significantly influences its perceived taste, with the breakdown of flesh during mastication affecting flavor release and mouthfeel [30]. Key textural attributes, including hardness, springiness, cohesiveness, chewiness, and resilience, were measured and presented in Figure 2. Notably, springiness and cohesiveness did not change significantly over three years, indicating that the grey jujube retained a certain degree of toughness. Hardness and chewiness values increased substantially within the first year of storage, by 39.0% and 47.4%, respectively, and remained stable thereafter. This increase suggests initial structural reinforcement, likely due to moisture loss, which affects cell turgor and tissue compactness. Additionally, resilience decreased by 15.3% over two years, further indicating moisture-related changes affecting texture. As illustrated by the texture profiles in Figure 2, the storage process had a pronounced effect on grey jujube textural properties, likely due to ongoing dehydration and associated cellular structure changes. These findings highlight the impact of long-term storage on grey jujube quality, emphasizing the role of moisture loss.

#### 3.1.3. Chemical Properties

As shown in Table 1, the moisture content of grey jujube fruit gradually decreased over the storage period, reaching 11.3 ± 0.3% (Y2) and 10.8 ± 0.3% (Y3) (*p* < 0.05). This reduction in moisture content likely results from water evaporation during storage, correlating with increased hardness and chewiness due to decreased cellular turgor. The association between moisture loss and increased textural firmness highlights the impact of storage on the fruit’s structural properties.

Sugar content, a key determinant of nutritional quality and flavor, exhibited notable changes during storage, as shown in Table 1. Reducing sugar content decreased significantly over time, from 42.0 ± 0.4% (Y0) to 39.7 ± 0.5% (Y1), whereas total sugar content exhibited only a slight reduction after the first year. In grey jujube, sucrose, glucose, fructose, and minor amounts of xylose constitute the primary sugars, with glucose, fructose, and xylose being the main reducing sugars. All three components decreased consistently throughout storage, with xylose showing a marked decline. The observed decrease in reducing sugars is consistent with the overall reduction trend and may be attributed to the fruit’s respiration and the Maillard reaction occurring during storage, processes known to consume sugars as precursors [30].

Total acidity levels significantly increased with storage time (Table 1), indicating a progressive accumulation of organic acids. This increase may result from the Maillard reaction, which causes amino acid loss and sugar degradation, leading to organic acid formation and increased acidity [13]. Furthermore, while respiration rates remain relatively low during storage, they may still play a role in the gradual acid accumulation by consuming sugars and releasing metabolic by-products, contributing to the observed increase in acidity.

Protein content remained stable throughout the storage period, indicating its resilience despite changes in other nutritional components.

#### 3.1.4. Antioxidant Components

Four bioactive compounds were monitored throughout storage: vitamin C, total polyphenols, total flavonoids, and cyclic adenosine monophosphate (cAMP), as shown in Figure 3. Results indicate a significant decline in vitamin C, total polyphenols, and cAMP content over the three-year storage period, while total flavonoids remained stable. Specifically, vitamin C, total polyphenols, and cAMP decreased by 36.9%, 21.2%, and 38.1%, respectively, suggesting susceptibility to oxidative degradation under prolonged storage conditions. The reduction in vitamin C and polyphenols content may result from oxidative reactions triggered by residual enzymatic activity and ambient oxygen, processes which can concurrently impact color stability, leading to a progressive darkening of the fruit. These oxidative losses not only reduce the antioxidant capacity of grey jujube but also affect its sensory qualities and nutritional profile, potentially diminishing the fruit’s perceived health benefits [31].

Notably, the stability of total flavonoids across the storage period contrasts with the decline observed in other antioxidants, suggesting that flavonoids in grey jujube may be more resistant to oxidation under cold storage. The increase in total flavonoid content after the first year may be due to the loss of substances such as sugars, which decreases the total dry matter content. Flavonoid stability could be advantageous in maintaining some level of antioxidant protection. Flavonoid stability during storage, particularly their resistance to oxidation, could be attributed to several factors. Flavonoids possess a characteristic C6-C3-C6 structure, consisting of two aromatic rings (A and B) connected by a three-carbon bridge (C). This arrangement allows flavonoids to stabilize free radicals through their conjugated double bonds, enhancing their antioxidant properties. The presence of hydroxyl groups on the aromatic rings further contributes to their ability to neutralize oxidative agents. Additionally, glycosylation can affect their stability and bioavailability. The structural stability of flavonoids, particularly their aromatic ring system and hydroxyl groups, makes them resistant to oxidative degradation, thus preserving their antioxidant activity during cold storage. However, the precise mechanisms of this stability, including potential interactions with other compounds in jujube, require further investigation. Future studies will further investigate this aspect to understand how specific flavonoid compounds contribute to antioxidant protection during storage.

#### 3.1.5. Changes of Free Amino Acids

A total of 17 amino acids were quantified in grey jujube samples during storage (Table 2), with most showing a gradual decline over time, except for proline, which remained stable. The trend in amino acid content changes during storage was clearly visible in the clustering heatmap (Figure 4). Proline was the most abundant amino acid, accounting for nearly half of the total amino acid content. Significant changes were observed in aspartic acid, glycine, alanine, phenylalanine, lysine, and histidine, which showed notable reductions, whereas threonine and methionine levels remained relatively stable over the two-year storage period. Other amino acids, including serine, glutamate, cystine, valine, isoleucine, leucine, tyrosine, and arginine, exhibited no significant change during the first year but declined significantly after two years of storage.

The stability of proline can be attributed to two primary factors. First, proline’s unique secondary amine structure makes it less prone to undergoing non-enzymatic browning reactions under the ambient temperatures typically employed during storage. This structural resilience likely minimizes proline’s involvement in such reactions, preserving its content over time. Second, although protein hydrolysis during storage could potentially generate proline, it seems that any contributions from this process are offset by concurrent degradation mechanisms, resulting in a relatively stable proline level throughout the storage period [13].

The observed reductions in other amino acids likely result from several storage-related factors. Amino acids with reactive side chains, such as lysine and arginine, are particularly prone to degradation through oxidative pathways, even at lower temperatures [32]. This degradation contributes not only to a reduction in amino acid content but also to the formation of compounds that may influence the fruit’s flavor, color, and nutritional profile. The gradual reduction in these amino acids may, therefore, impact both the organoleptic properties and the nutritional value of stored grey jujube, underscoring the potential quality decline over extended storage periods.

In summary, the stability of proline coupled with the degradation of other amino acids highlights the complex interplay of chemical reactions during storage, with implications for both the sensory quality and nutritional composition of grey jujube. These findings suggest that optimizing storage conditions to limit oxidative reaction activity could help preserve amino acid integrity, particularly for amino acids vulnerable to degradation. Further research focusing on storage interventions that minimize amino acid loss could enhance the overall quality and nutritional retention in long-term stored grey jujube products.

### 3.2. Sensory Evaluation

The results of the sensory evaluation of grey jujubes are shown in the radar charts in Figure 5. Sensory evaluation of grey jujubes over the storage period revealed significant changes across several key flavor attributes. The grey jujube fragrance score decreased markedly, from 4.2 in control sample (Y0) to 0.8 after three years (Y3), indicating a substantial loss of fresh aroma with prolonged storage. Meanwhile, the rancid flavor score increased from 0.3 in Y0 to 2.5 in Y3, suggesting the development of off-flavors associated with lipid oxidation and other degradative processes over time. Sourness showed a slight increase from 1.5 in Y0 to 2 in Y3, while sweetness declined from 3.7 in Y0 to 2.4 in Y3, indicating a shift toward a less desirable flavor profile. Bitterness scores also rose modestly, from 0.1 in Y0 to 0.8 in Y3, and aftertaste scores decreased from 3.1 in Y0 to 1.0 in Y3, reflecting a reduction in the persistence of pleasant flavors that contribute to overall eating satisfaction.

These sensory results align closely with the findings from chemical analyses, which showed a decrease in volatile compounds associated with fresh and sweet flavors, such as specific alcohols and aldehydes, and an increase in compounds contributing to complex, and sometimes off-putting, flavors, such as esters and organic acids. The reduction in sweetness and increase in sourness and bitterness can be attributed to the breakdown of sugars and the accumulation of organic acids as storage progresses. This shift in taste profile is commonly observed in fruit ripening and senescence processes, where enzymatic activities and metabolic changes alter the fruit’s chemical composition [33,34].

Furthermore, the decline in aftertaste scores suggests that the degradation of flavor compounds associated with long-lasting sensory impressions plays a role in diminishing the overall flavor experience of stored grey jujube. These findings emphasize that extended storage not only impacts grey jujube’s chemical profile but also directly influences its sensory qualities, potentially affecting consumer acceptability. To mitigate these effects, storage conditions such as temperature and humidity control could be optimized to preserve the freshness and overall flavor profile, thereby extending shelf life while maintaining product quality.

### 3.3. Volatile Aroma Compounds

#### 3.3.1. Profile of Grey Jujube Flavor in Different Storage Periods

The sensory quality of grey jujube is closely linked to its volatile profile, which undergoes significant changes during storage. To investigate these changes, the volatile compounds in grey jujube during the storage period (Y0–Y3) were analyzed. By application of GC-MS, the aroma compounds in the grey jujube samples were characterized in comparison of the RIs and mass spectra. The concentrations, retention indices (RI) and odor activity values (OAVs) of these aroma compounds are summarized in Table 2, which lists 70, 70, 61, 64, 64, 63 and 63 compounds corresponding to the Y0, Y0.5, Y1, Y1.5, Y2, Y2.5 and Y3 grey jujube samples, respectively. Among the total compounds, there were 13 aldehydes, 7 alcohols, 27 esters, 9 ketones and 15 acids, which were the most abundant compounds in grey jujube.

Among the volatiles, esters accounted for the highest semiquantitative volatile portion (Figure 6), with 21 esters were identified in all samples, regarded as one of the most important flavoring compounds in control sample. Pentanoic acid, ethyl ester (46), 2-Propenoic acid, 1-methylundecyl ester (50) were only detected in the Y0 and Y0.5 samples, and methyl formate (53) was only detected in the Y0, Y0.5 and Y1 samples. The second highest semiquantitative volatile portion consisted of acids in the Y2-Y3 samples (Figure 6). A total of 15 acids were identified, the main components of which were acetic acid (21), hexanoic acid (27) and n-decanoic acid (33). The most quantity-predominant compound in Y2-Y3 was acetic acid (77.6 μg/kg in the Y2, 116.09 μg/kg in the Y3 and 122.2 μg/kg in the Y3).

The ester and acid contents in this study were consistent with previously reported quantification studies of grey jujube [35]. Alcohols and aldehydes were also the main components of the volatile compounds in dates, which were known to impart a fresh and green odor to grey jujube. The storage of grey jujube leads to significant changes in profile, with a marked reduction in alcohols, aldehydes, and esters, which were responsible for fresh, fruity, and sweet aromas. Concurrently, the increase in acids, particularly acetic acid, indicates a shift toward sourness and off-flavor development. These findings align with previous studies on the postharvest deterioration of fruit quality during prolonged storage.

**Table 2 foods-14-00050-t002:** Volatile compounds’ concentration (µg/kg) and their odor activity values (OAVs) in seven grey jujube samples.

**NO**	**Compound Name ^A^**	**CAS**	RI ^B^	LRI ^C^	Identification ^D^	Concentration (Mean ± SD) (μg/kg)							OAV						
						Y0	Y0.5	Y1	Y1.5	Y2	Y2.5	Y3	Y0	Y0.5	Y1	Y1.5	Y2	Y2.5	Y3
Alcohols																			
1	2-Octen-1-ol	22104-78-5	1292	- ^E^	MS, RI	25.4 ± 3.5 a ^G^	22.1 ± 0.51 b	17.5 ± 1.3 c	8.8 ± 1.14 d	9.20 ± 1.0 d	9.98 ± 1.39 d	10.5 ± 0.8 d	0.509	0.443	0.35	0.176	0.184	0.200	0.211
2	1-Octen-3-ol	3391-86-4	1452	1441	MS, RI	71.2 ± 3.2 a	61.23 ± 1.23 b	57.6 ± 2.5 b	41.61 ± 5.96 c	42.9 ± 3.8 c	31.05 ± 4.33 d	31.9 ± 2.6 d	71.200	61.230	57.6	41.610	42.9	31.050	31.9
3	2-Propyl-1-pentanol	58175-57-8	1491	-	MS, RI	48.3 ± 3.8 a	42.83 ± 0.56 b	26.6 ± 1.9 c	25.62 ± 4.26 c	26.5 ± 2.6 c	26.79 ± 4.11 c	28.4 ± 3.5 c	/ ^F^	/	/	/	/	/	/
4	2-Decen-1-ol	56030-49-0	1501	-	MS, RI	9.59 ± 0.7 a	8.28 ± 0.15 b	6.90 ± 0.3 c	6.26 ± 0.95 cd	6.5 ± 0.5 cd	5.45 ± 0.86 d	5.70 ± 0.4 d	/	/	/	/	/	/	/
5	Benzyl alcohol	100-51-6	1877	-	MS, RI	3.49 ± 0.41 a	3.06 ± 0.02 b	2.99 ± 0.23 b	0 c	0 c	0 c	0 c	0.001	0.001	0.0005	0.000	0	0.000	0
6	Phenylethyl Alcohol	1960/12/8	1911	-	MS, RI	1.56 ± 0.11 b	1.37 ± 0.03 b	1.04 ± 0.06 c	1.46 ± 0.21 b	1.54 ± 0.08 b	1.9 ± 0.31 a	2.00 ± 0.15 a	0.035	0.030	0.0232	0.032	0.0341	0.042	0.0445
7	Phenol, 4-(3-hydroxy-1-propenyl)-2-methoxy-	458-35-5	2148	-	MS, RI	1.09 ± 0.11 a	0.94 ± 0.01 b	0 c	0 c	0 c	0 c	0 c	/	/	/	/	/	/	/
Aldehydes													4.260	3.719	3.66	3.097	3.26	0.606	0.643
8	Hexanal	66-25-1	1085	1081	MS, RI	15.6 ± 1.2 a	13.62 ± 0.32 b	13.4 ± 0.8 b	11.34 ± 1.62 c	11.9 ± 2.2 c	2.22 ± 0.33 d	2.35 ± 0.15 d	2.020	1.778	0	0.000	0	0.000	0
9	Heptanal	111-71-7	1189	1180	MS, RI	10.1 a	8.89 ± 0.1 b	0 c	0 c	0 c	0 c	0 c	0.128	0.112	0.114	0.065	0.0687	0.020	0.021
10	2-Hexenal	505-57-7	1222	-	MS, RI	8.58 ± 0.75 a	7.52 ± 0.05 b	7.61 ± 0.62 b	4.35 ± 0.65 c	4.6 ± 0.54 c	1.35 ± 0.21 d	1.41 ± 0.31 d	1.140	0.999	1.04	0.840	0.89	0.289	0.301
11	2-Heptenal, (*Z*)-	57266-86-1	1327	1291	MS, RI	11.4 ± 1.5 a	9.99 ± 0.29 ab	10.4 ± 0.09 a	8.4 ± 1.28 c	8.9 ± 0.67 bc	2.89 ± 0.37 d	3.01 ± 0.52 d	5.970	5.293	3.7	3.482	3.63	3.428	3.56
12	Nonanal	124-19-6	1396	1392	MS, RI	20.9 ± 1.8 a	18.53 ± 0.48 b	12.9 ± 1.2 c	12.19 ± 1.65 c	12.7 ± 0.9 c	12 ± 1.96 c	12.5 ± 0.8 c	113.000	97.200	103	43.700	45.5	25.200	26.5
13	2-Octenal, (*E*)-	2548-87-0	1433	-	MS, RI	22.6 ± 2.5 a	19.44 ± 0 b	20.6 ± 1.7 b	8.74 ± 1.27 c	9.10 ± 1.1 c	5.04 ± 0.69 d	5.31 ± 0.62 d	0.000	0.000	0	0.000	0	0.273	0.288
14	Furfural	1998/1/1	1470	1466	MS, RI	0 b	0 b	0 b	0 b	0 b	2.21 ± 0.33 a	2.33 a	0.805	0.700	0.566	0.520	0.556	0.474	0.494
15	Benzaldehyde	100-52-7	1527	1538	MS, RI	40.3 ± 3.2 a	35.06 ± 0.81 b	28.3 ± 2.3 c	26.04 ± 3.93 c	27.8 ± 3.1 c	23.71 ± 3.43 c	24.7 ± 1.9 c	6.420	5.603	4.21	3.368	3.5	2.768	2.93
16	2-Nonenal, (*Z*)-	60784-31-8	1538	-	MS, RI	3.85 ± 0.28 a	3.36 ± 0.02 b	2.52 ± 0.32 c	2.02 ± 0.3 cd	2.1 ± 0.17 cd	1.66 ± 0.24 d	1.76 ± 0.20 d	0.042	0.036	0.0315	0.059	0.0613	0.066	0.068
17	2-Furancarboxaldehyde, 5-methyl-	620-02-0	1578	1570	MS, RI	2.10 ± 0.19 b	1.81 ± 0.03 bc	1.58 ± 0.13 c	2.95 ± 0.4 a	3.07 ± 0.38 a	3.28 ± 0.49 a	3.4 ± 0.45 a	9.000	7.890	0	7.880	8.24	0.000	0
18	2-Decenal, (E)-	3913-81-3	1646	-	MS, RI	9.00 ± 0.85 a	7.89 ± 0.23 b	0 c	7.88 ± 1.17 b	8.24 ± 0.54 b	0 c	0 c	0.483	0.426	0.34	0.330	0.345	0.000	0
19	2-Undecenal, E-	53448-07-0	1753	-	MS, RI	1.45 ± 0.25 a	1.28 ± 0.02 b	1.02 ± 0.13 c	0.99 ± 0.15 c	1.03 ± 0.21 c	0 d	0 d	0.000	0.000	0	0.000	0	0.000	0
20	5-Hydroxymethylfurfural	67-47-0	2490	2510	MS, RI	0 b	0 b	0 b	0 b	0 b	0.47 ± 0.07 a	0.491 a	1.410	1.227	3.64	7.450	7.76	11.609	12.22
Acids													0.001	0.000	0.0006	0.001	0.0006	0.001	0.0005
21	Acetic acid	64-19-7	1463	1442	MS, RI	14.1 ± 2.1 d	12.27 ± 0.24 d	36.4 ± 4.2 c	74.5 ± 11.48 b	77.6 ± 5.8 b	116.09 ± 17.11 a	122.2 ± 8.7 a	/	/	/	/	/	/	/
22	Propanoic acid	1979/9/4	1550	1531	MS, RI	1.53 ± 0.22 bc	1.34 ± 0.03 c	1.90 ± 0.21 a	1.71 ± 0.24 ab	1.8 ± 0.17 ab	1.58 ± 0.25 abc	1.64 ± 0.12 abc	0.326	0.284	0.363	0.269	0.28	0.214	0.225
23	8-Benzoyloctanoic acid	16269-05-9	1653	-	MS, RI	0 c	0 c	0 c	0.48 ± 0.07 b	0.503 ± 0.062 b	0.79 ± 0.12 a	0.821 ± 0.076 a	0.002	0.002	0.0014	0.001	0.0014	0.001	0.0013
24	Pentanoic acid, 4-methyl-	646-07-1	1679	-	MS, RI	3.26 ± 0.30 a	2.84 ± 0.07 ab	3.63 ± 0.29 a	2.69 ± 0.42 bc	2.8 ± 0.19 ab	2.14 ± 0.35 d	2.25 ± 0.17 cd	/	/	/	/	/	/	/
25	Dodecanoic acid, 3-hydroxy-	1883-13-2	1711	-	MS, RI	0.965 ± 0.085 a	0.84 ± 0.02 b	0.690 ± 0.072 c	0.65 ± 0.1 c	0.687 ± 0.065 c	0.64 ± 0.1 c	0.672 ± 0.052 c	0.202	0.175	0.167	0.132	0.137	0.118	0.121
26	Propanedioic acid, propyl-	616-62-6	1747	-	MS, RI	3.75 ± 0.41 a	3.29 ± 0.06 b	2.41 ± 0.17 c	2.22 ± 0.27 c	2.3 ± 0.22 c	2.04 ± 0.29 c	2.16 ± 0.19 c	/	/	/	/	/	/	/
27	Hexanoic acid	142-62-1	1853	1855	MS, RI	40.4 ± 3.6 a	35.01 ± 0.23 b	33.4 ± 2.9 b	26.4 ± 4.43 c	27.4 ± 3.2 c	23.57 ± 3.82 c	24.3 ± 2.4 c	/	/	/	/	/	/	/
28	cis-5-Dodecenoic acid	2430-94-6	1897	-	MS, RI	1.18 ± 0.15 ab	1.02 ± 0.02 ab	0.961 ± 0.085 b	1.2 ± 0.2 a	1.25 ± 0.16 a	1.02 ± 0.17 ab	1.08 ± 0.13 ab	0.069	0.060	0.0498	0.036	0.037	0.031	0.032
29	Heptanoic acid, 2-ethyl-	3274-29-1	1956	-	MS, RI	0.901 ± 0.085 a	0.78 ± 0.02 a	0.745 ± 0.094 a	0.85 ± 0.14 a	0.900 ± 0.093 a	0.76 ± 0.12 a	0.797 ± 0.185 a	0.022	0.019	0.0151	0.013	0.0132	0.010	0.0105
30	Heptanoic acid	111-14-8	1961	1948	MS, RI	6.86 ± 0.58 a	5.97 ± 0.07 b	4.98 ± 0.45 c	3.56 ± 0.45 d	3.71 ± 0.22 d	3.13 ± 0.48 d	3.20 ± 0.29 d	0.003	0.002	0.0012	0.001	0.001	0.001	0.0009
31	Octanoic acid	124-07-2	2068	-	MS, RI	11.0 ± 0.9 bc	9.42 ± 0.13 a	7.54 ± 0.82 b	6.4 ± 0.94 c	6.62 ± 0.65 bc	5.07 ± 0.82 d	5.23 ± 0.62 d	0.003	0.002	0.0023	0.002	0.0019	0.001	0.0013
32	Nonanoic acid	112-05-0	2155	2177	MS, RI	14.0 ± 1.1 a	12.18 ± 0.14 b	6.16 ± 0.75 c	4.93 ± 0.74 cd	5.12 ± 0.26 cd	4.39 ± 0.74 d	4.64 ± 0.53 d	/	/	/	/	/	/	/
33	n-Decanoic acid	334-48-5	2258	-	MS, RI	28.2 ± 2.1 a	24.72 ± 0.16 b	23.4 ± 1.7 b	18.24 ± 2.84 c	19.2 ± 2.1 c	12.77 ± 1.82 d	13.4 ± 1.2 d	0.027	0.024	0.025	0.016	0.0166	0.011	0.011
34	Palmitoleic acid	373-49-9	2279	-	MS, RI	3.56 ± 0.41 a	3.14 ± 0.05 b	2.4 ± 0.22 c	1.57 ± 0.24 d	1.62 ± 0.21 d	1.41 ± 0.2 d	1.45 ± 0.19 d	0.019	0.016	0.0166	0.009	0.00893	0.000	0
35	Dodecanoic acid	143-07-7	2381	2375	MS, RI	13.5 ± 1.4 a	11.84 ± 0.21 a	12.5 ± 1.1 a	8.03 ± 1.03 b	8.31 ± 0.91 b	5.31 ± 0.7 c	5.51 ± 0.43 c	2017.000	1746.955	1807	1516.917	1600	956.825	1008
Ketones													0.077	0.068	0.061	0.049	0.052	0.035	0.0361
36	3-Octanone	106-68-3	1257	-	MS, RI	18.7 ± 2.5 a	16.33 ± 0.22 b	16.6 ± 1.8 b	8.6 ± 1.29 c	8.93 ± 0.92 c	0 d	0 d	/	/	/	/	/	/	/
37	1-Octen-3-one	4312-99-6	1304	1295	MS, RI	6.05 ± 0.52 a	5.24 ± 0.09 bc	5.42 ± 0.43 ab	4.55 ± 0.68 c	4.81 ± 0.45 bc	2.87 ± 0.43 d	3.03 ± 0.18 d	/	/	/	/	/	/	/
38	5-Hepten-2-one, 6-methyl-	110-93-0	1340	1339	MS, RI	7.69 ± 0.62 a	6.77 ± 0.15 ab	6.10 ± 0.57 bc	4.92 ± 0.77 d	5.23 ± 0.49 cd	3.5 ± 0.5 e	3.61 ± 0.29 e	0.357	0.311	0	0.000	0	0.000	0
39	Cyclohexanone, 2-(hydroxymethyl)-	5331/8/8	1382	-	MS, RI	0.706 a	0.62 ± 0.01 b	0 c	0 c	0 c	0 c	0 c	/	/	/	/	/	/	/
40	3-tert-Butyl-2-pyrazolin-5-one	29211-68-5	1565	-	MS, RI	2.14 ± 0.12 a	1.84 ± 0.02 b	0 d	1.93 ± 0.26 ab	1.99 ± 0.13 ab	1.25 ± 0.21 c	1.31 ± 0.09 c	/	/	/	/	/	/	/
41	2-Undecanone	112-12-9	1600	-	MS, RI	3.57 a	3.11 ± 0.04 b	0 c	0 c	0 c	0 c	0 c	/	/	/	/	/	/	/
42	2-Undecanone, 6,10-dimethyl-	1604-34-8	1689	-	MS, RI	4.25 ± 0.52 a	3.71 ± 0.05 b	3.45 ± 0.38 b	2.57 ± 0.4 cd	2.70.12 c	2.08 ± 0.28 e	2.16 ± 0.24 de	9.460	8.059	10	10.861	11.5	0.000	0
43	(*S*)-(+)-2′,3′-Dideoxyribonolactone	32780-06-6	1704	-	MS, RI	4.22 ± 0.35 a	3.71 ± 0.11 b	2.77 ± 0.24 c	2.39 ± 0.35 cd	2.52 ± 0.31 cd	2.07 ± 0.34 d	2.14 ± 0.25 d	0.013	0.011	0	0.000	0	0.000	0
44	2-Pentadecanone, 6,10,14-trimethyl-	502-69-2	2078	-	MS, RI	1.67 ± 0.12 a	1.44 ± 0.02 b	1.03 ± 0.09 c	0.92 ± 0.16 c	0.956 ± 0.07 c	0.69 ± 0.12 d	0.722 ± 0.08 d	77.600	68.280	48.1	24.360	25.1	21.660	22.4
esters													/	/	/	/	/	/	/
45	Butanoic acid, 3-methyl-, ethyl ester	108-64-5	1137	-	MS, RI	1.89 ± 0.25 bc	1.61 ± 0.01 c	2.0 ± 0.17 ab	2.17 ± 0.33 ab	2.30 ± 0.21 a	0 d	0 d	0.012	0.010	0.0114	0.011	0.0117	0.010	0.0106
46	Pentanoic acid, ethyl ester	539-82-2	1140	1145	MS, RI	1.18 a	1.02 ± 0.02 b	0 c	0 c	0 c	0 c	0 c	/	/	/	/	/	/	/
47	Pentanoic acid, 4-methyl-, ethyl ester	25415-67-2	1236	1242	MS, RI	38.8 ± 4.5 a	34.14 ± 0.39 b	24.1 ± 3.2 c	12.18 ± 1.79 d	12.6 ± 0.9 d	10.83 ± 1.68 d	11.2 ± 0.7 d	/	/	/	/	/	/	/
48	Hexanoic acid, 2-phenylethyl ester	6290-37-5	1261	-	MS, RI	0 b	0 b	0 b	3.64 ± 0.51 a	3.85 a	0 b	0 b	0.004	0.004	0.00199	0.002	0.0018	0.002	0.00193
49	Heptanoic acid, ethyl ester	106-30-9	1336	-	MS, RI	2.00 ± 0.16 a	1.71 ± 0.02 a	1.94 ± 0.09 a	1.88 ± 0.28 a	1.99 ± 0.13 a	1.71 ± 0.29 a	1.81 ± 0.11 a	0.000	0.000	0.0001	0.000	0	0.000	0
50	2-Propenoic acid, 1-methylundecyl ester	51443-73-3	1348	-	MS, RI	1.28 a	1.12 ± 0.03 b	0 c	0 c	0 c	0 c	0 c	0.022	0.019	0.0221	0.018	0.019	0.016	0.0167
51	3-Methylheptyl acetate	72218-58-7	1386	-	MS, RI	3.19 ± 0.16 a	2.82 ± 0.02 b	2.01 ± 0.21 c	1.62 ± 0.22 d	1.71 ± 0.09 d	1.36 ± 0.18 e	1.39 ± 0.13 e	0.386	0.332	0.329	0.357	0.357	0.393	0.417
52	Octanoic acid, methyl ester	111-11-5	1391	1385	MS, RI	2.20 ± 0.12 a	1.94 ± 0.01 b	0.997 ± 0.10 c	0.87 ± 0.13 c	0.900 ± 0.08 c	0.94 ± 0.14 c	0.963 ± 0.08 c	1.230	1.054	1.11	1.083	1.12	1.073	1.11
53	Methyl formate	107-31-3	1464	-	MS, RI	20.5 ± 2.1 a	17.97 ± 0.43 b	16.4 ± 1.2 c	0 d	0 d	0 d	0 d	0.000	0.000	0	0.000	0	0.000	0
54	Decanoic acid, methyl ester	110-42-9	1596	1590	MS, RI	22.2 ± 1.6 a	19.46 ± 0.34 ab	22.1 ± 1.2 a	18.24 ± 2.99 bc	19.0 ± 0.8 abc	15.81 ± 2.36 c	16.7 ± 0.6 bc	0.000	0.000	0	0.000	0	0.004	0.0045
55	Benzoic acid, methyl ester	93-58-3	1624	-	MS, RI	1.08 ± 0.09 ab	0.93 ± 0 b	0.921 ± 0.103 b	1 ± 0.16 ab	1.05 ± 0.08 ab	1.1 ± 0.16 ab	1.17 ± 0.12 a	/	/	/	/	/	/	/
56	Decanoic acid, ethyl ester	110-38-3	1640	1636	MS, RI	24.6 ± 2.2 a	21.07 ± 0.28 a	22.2 ± 1.9 a	21.65 ± 3.37 a	22.4 ± 0.8 a	21.46 ± 2.76 a	22.2 ± 1.2 a	0.010	0.008	0.0079	0.008	0.0085	0.008	0.0087
57	Benzoic acid, ethyl ester	93-89-0	1668	-	MS, RI	7.32 ± 0.51 ab	6.32 ± 0.11 b	7.72 ± 0.24 a	7.09 ± 1.09 ab	7.52 ± 0.37 ab	7.49 ± 1.17 ab	7.75 ± 0.39 a	2.830	2.461	2.61	1.993	2.08	2.074	2.14
58	Benzeneacetic acid, ethyl ester	101-97-3	1788	-	MS, RI	0 b	0 b	0 b	0 b	0 b	0.43 ± 0.06 b	0.452 b	1.930	1.670	1.54	1.020	1.08	0.940	0.99
59	Dodecanoic acid, methyl ester	111-82-0	1802	1814	MS, RI	21.1 ± 1.2 a	18.29 ± 0.32 b	16.7 ± 1.2 b	12.34 ± 1.94 c	12.9 ± 0.8 c	9.17 ± 1.19 d	9.59 ± 0.5 d	/	/	/	/	/	/	/
60	Dodecanoic acid, ethyl ester	106-33-2	1844	-	MS, RI	33.1 ± 2.1 a	29.02 ± 0.38 a	27.7 ± 2.6 a	28.21 ± 4.08 a	29.9 ± 1.4 a	29.38 ± 4.94 a	30.5 ± 1.9 a	0.026	0.023	0.0215	0.019	0.019	0.016	0.0169
61	Benzenepropanoic acid, ethyl ester	2021-28-5	1885	-	MS, RI	4.53 ± 0.35 a	3.94 ± 0.09 ab	4.18 ± 0.29 a	3.19 ± 0.44 c	3.33 ± 0.33 c	3.32 ± 0.47 c	3.42 ± 0.28 bc	/	/	/	/	/	/	/
62	2(3H)-Furanone, 5-hexyldihydro-	706-14-9	1917	1905	MS, RI	1.93 ± 0.21 a	1.67 ± 0.02 b	1.54 ± 0.13 b	1.02 ± 0.15 c	1.08 ± 0.09 c	0.94 ± 0.14 c	0.988 ± 0.11 c	27.700	23.862	15.5	23.528	24.4	18.856	19.7
63	Methyl tetradecanoate	124-10-7	2009	-	MS, RI	1.91 ± 0.16 ab	1.64 ± 0.03 bc	1.42 ± 0.11 c	1.89 ± 0.3 ab	1.95 ± 0.09 a	0.99 ± 0.14 d	1.04 ± 0.17 d	1.040	0.920	0.808	0.780	0.806	0.610	0.643
64	ç-Dodecalactone	2305/5/7	2031	-	MS, RI	1.63 ± 0.21 a	1.43 ± 0.04 ab	1.35 ± 0.13 bc	1.19 ± 0.17 bd	1.21 ± 0.11 bd	1.01 ± 0.17 d	1.07 ± 0.12 d	/	/	/	/	/	/	/
65	Tetradecanoic acid, ethyl ester	124-06-1	2051	-	MS, RI	7.39 ± 0.53 a	6.38 ± 0.09 b	5.46 ± 0.42 c	4.73 ± 0.69 cd	4.93 ± 0.37 cd	4.34 ± 0.61 d	4.46 ± 0.32 d	/	/	/	/	/	/	/
66	2-Propenoic acid, 3-phenyl-, ethyl ester	103-36-6	2127	-	MS, RI	1.66 ± 0.21 a	1.43 ± 0.03 b	0.928 ± 0.0714 d	1.41 ± 0.22 b	1.47 ± 0.15 ab	1.13 ± 0.17 cd	1.18 ± 0.10 c	/	/	/	/	/	/	/
67	2(3H)-Furanone, 5-hexyldihydro-	706-14-9	2132	-	MS, RI	1.04 ± 0.11 a	0.92 ± 0.01 ab	0.808 ± 0.091 bc	0.78 ± 0.1 c	0.806 ± 0.082 bc	0.61 ± 0.09 d	0.643 ± 0.071 d	/	/	/	/	/	/	/
68	7-Hexadecenoic acid, methyl ester, (Z)-	56875-67-3	2174	-	MS, RI	0.742 ± 0.063 b	0.65 ± 0.02 b	0.656 ± 0.041 b	0.9 ± 0.13 a	0.916 ± 0.102 a	0.46 ± 0.08 c	0.487 ± 0.058 c	0.063	0.055	0.0732	0.058	0.06	0.060	0.063
69	Ethyl 9-hexadecenoate	54546-22-4	2245	-	MS, RI	1.41 ± 0.13 a	1.23 ± 0.02 b	1.2 ± 0.05 b	1.06 ± 0.15 bc	1.12 ± 0.08 bc	0.93 ± 0.14 c	0.962 ± 0.053 c	/	/	/	/	/	/	/
70	2(4H)-Benzofuranone, 5,6,7,7a-tetrahydro-4,4,7a-trimethyl-	15356-74-8	2333	-	MS, RI	0.993 ± 0.051 a	0.88 ± 0.01 a	0.870 ± 0.081 a	0.69 ± 0.11 b	0.726 ± 0.063 b	0.51 ± 0.08 c	0.529 ± 0.071 c	/	/	/	/	/	/	/
71	Octaethylene glycol monododecyl ether	3055-98-9	2350	-	MS, RI	0.746 ± 0.039 bc	0.64 ± 0 bd	0.920 ± 0.085 a	0.84 ± 0.14 ab	0.875 ± 0.076 ab	0.5 ± 0.07 e	0.529 ± 0.042 de	/	/	/	/	/	/	/
Others													/	/	/	/	/	/	/
72	D-Limonene	5989-27-5	1198	-	MS, RI	2.15 ± 0.09 ab	1.88 ± 0.03 b	2.49 ± 0.19 a	1.97 ± 0.28 b	2.05 ± 0.25 b	2.03 ± 0.31 b	2.13 ± 0.23 ab	0.0632		0.0732		0.06		0.063
73	Cyclohexene, 1-methyl-5-(1-methylethenyl)-, (R)-	1461-27-4	1199	-	MS, RI	54.6 a	47.32 ± 0.83 b	0 c	0 c	0 c	0 c	0 c	/		/		/		/
74	1H-Indene, 1-methylene-	2471-84-3	1740	-	MS, RI	3.79 ± 0.41 a	3.35 ± 0.04 b	2.33 ± 0.19 c	2.29 ± 0.36 c	2.41 ± 0.12 c	2.29 ± 0.33 c	2.43 ± 0.17 c	/		/		/		/
75	Phenol, 2,4-bis(1,1-dimethylethyl)-	96-76-4	2294	-	MS, RI	1.16 ± 0.08 a	1.01 ± 0.02 bc	1.11 ± 0.10 ab	0.88 ± 0.15 c	0.918 ± 0.067 c	0.68 ± 0.09 d	0.707 ± 0.058 d	/		/		/		/
76	N-Aminopyrrolidine	16596-41-1	2315	-	MS, RI	0 b	0 b	0 b	0 b	0 b	0.69 ± 0.12 a	0.725 a	/		/		/		/

^A^ The aroma compounds identified on the TG-WAXMS column. ^B^ The retention index of volatile compounds on the TG-WAXMS column. ^C^ The retention index of volatile compounds from published literature and online library. ^D^ RI: retention index; MS: mass spectrometry [36]. ^E^ The aroma compounds not identified in published literature and online library [37]. ^F^ The threshold of volatile compounds not found. ^G^ Values with different superscript Roman letters (a–e) in the same row are significantly different according to the Duncan test (*p* < 0.05). (Y0) The pre-storage control sample; (Y0.5) samples stored for six months; (Y1) samples stored for one year; (Y1.5) samples stored for one and a half years; (Y2) samples stored for two years; (Y2.5) samples stored for two and a half years; (Y3) samples stored for three years.

#### 3.3.2. Changes of Aroma-Active Compounds During the Grey Jujube Storage Process

OAVs have been widely used to assess the aroma potency of foods by considering the balance between the food matrix and the surrounding air, as reported in previous studies [38,39]. Odor Activity Values (OAVs) greater than 1 indicate that specific compounds significantly influence the sensory attributes of grey jujubes. In this study, the OAVs of 49 volatile compounds with documented odor threshold values, as shown in Table 2, were calculated to evaluate their contributions to the overall aroma of grey jujubes. Twenty-seven volatile compounds were excluded from the analysis due to insufficient odor threshold data. Among the remaining compounds, 18 were identified as aroma-active based on their OAV values, making a relatively high contribution to the overall aroma. From the aroma-active Compounds, there were seven aldehydes, one alcohol, seven esters, one ketone and one acid. Hierarchical clustering, including heat maps (Figure 7), was performed based on the changes in the identified aroma compounds to illustrate the aroma profile of the seven grey jujube samples with different storage periods. As storage time increased, there were significant changes in the content of key aroma-active compounds in the samples.

Octen-3-ol (2), also known as Moguol, was the only alcohol among the key aroma substance of grey jujube. However, the concentration of the alcohol significantly decreased after one year of storage (Y1) to 51.6 µg/kg, respectively, and further declined in Y3 (31.9 µg/kg), indicating a loss of freshness in the flavor profile with prolonged storage. The reduction in alcohols is often associated with enzymatic degradation and the reduction in green and fresh notes over time.

Seven aroma-active aldehydes were identified in grey jujube, including hexanal (8), heptanal (9), (Z)-2-heptenal (11), nonanal (12), (E)-2-octenal (13), (Z)-2-nonenal (16), and (E)-2-decenal (18). These compounds were commonly associated with “green”, “cut grass”, “fat”, and “citrus” notes, which were also noted by panelists in the sensory descriptions of grey jujube and were typical of many other jujube varieties [40,41]. During storage, the aldehydes were decreased to varying degrees. The decrease in aldehydes indicates a shift from fruity to more stale or off-flavor notes as the storage period increases. This trend reflects the oxidation and polymerization of aldehydes, which are commonly observed in the postharvest storage of fruits.

Seven aroma-active esters were identified in grey jujube, including pentanoic acid, 4-methyl-, ethyl ester (47), butanoic acid, 3-methyl-, ethyl ester (45), and others. Esters, key contributors to fruity and sweet aromas, were substantially diminished during storage. For instance, Pentanoic acid, 4-methyl-, ethyl ester (47) volatile in Y0, was present at 38.8 µg/kg, but this concentration dropped drastically to 24.1 µg/kg in Y1 and further declined to 11.2 µg/kg in Y3. The reduction in esters is typically associated with hydrolysis and oxidation reactions that occur during storage, leading to a loss of the fruity and sweet aromas characteristic of fruit [42].

Acetic acid was the only acid in the key aroma substances of grey jujube, and also the only substance whose content was significantly increased in the storage process. In Y0, acetic acid (21) was measured at 14.1 µg/kg, but after one year (Y1), its concentration increased sharply to 36.4 µg/kg, and then to 122.2 µg/kg in Y3, suggesting a shift toward sourness and pungency. Acetic acid (21) is a well-known marker of spoilage or undesirable fermentation processes during storage. This increase in acid concentration may be due to activity or oxidative degradation of sugars and fatty acids [43].

The results suggest that prolonged storage significantly alters the volatile profile of dates, with a marked reduction in desirable aroma compounds alongside an increase in pungent acids. This finding underlines the importance of optimizing storage conditions to retain the sensory quality of dates. These insights can provide valuable guidance for the date industry to enhance product quality and consumer satisfaction.

### 3.4. Principal Partial Least Square Regression (PLSR)

To elucidate the correlation between the physicochemical constituents, key aroma-active compounds, and sensory attributes of grey jujube, partial least squares regression (PLSR) analysis was conducted. The compounds selected for analysis were rigorously chosen based on their relevance to the study’s objectives, ensuring they represent critical influences on sensory qualities. Specifically, nine conventional physicochemical parameters (moisture content, total sugar, reducing sugar, protein, total acid, Xylose, fructose, glucose, and sucrose), four antioxidant-active components (vitamin C, total polyphenols, total flavonoids, and cyclic adenosine monophosphate), 17 amino acids (threonine, serine, glutamate, glycine, alanine, cystine, valine, methionine, isoleucine, leucine, tyrosine, phenylalanine, lysine, histidine, arginine, and proline), and 17 key aroma compounds (listed in Figure 7) were incorporated as predictor variables in the PLSR model. Here, the X-matrix encompassed the chemical composition of the grey jujube samples, while the Y-matrix corresponded to the sensory attributes of interest. The developed PLS2 model comprised three principal components, accounting for 100% of the cross-validated variance, indicating the model’s effectiveness in accurately capturing the comprehensive information of the samples. The first two principal components (PC1 and PC2) were sufficient to represent the overall sample information, explaining 98.7% of the cross-validated variance, while additional components (PCs) contributed minimal useful information. Consequently, only the correlation loading plots for PC1 and PC2 are presented (Figure 8). As shown in Figure 8, nearly all variables fall in the two ellipses (R^2^ = 0.5 and R^2^ = 1.0), representing 50% and 100% of the explained variance, respectively, indicating that these variables are well-explained by the PLSR model. Furthermore, the 47 compounds significantly impacted one or more of the five sensory descriptors.

The PLSR analysis indicates that the seven samples (Y0, Y0.5, Y1, Y1.5, Y2, Y2.5, Y3) are distinctly separated in the correlation loading plot, highlighting significant variations in their physicochemical and aroma profiles. On the primary component, PC1, sensory attributes such as grey jujube fragrance and sweet, along with the majority of physicochemical and aroma-active compounds, are positioned on the right side of the loading plot, predominantly outside the 50% explained variance ellipse. This arrangement suggests a robust positive correlation between most physicochemical and aroma compounds with the sensory attributes grey jujube fragrance and sweet, while a strong inverse correlation with bitter, rancid, and sour is also observed. The decline in these compounds during storage is associated with a progressive attenuation of the sensory qualities grey jujube fragrance and sweet, underscoring the role of specific compounds in preserving favorable sensory traits. Conversely, attributes such as bitter, rancid, and sour, along with compounds like proline, protein, total acid, total flavonoids, and acetic acid (21), are located on the left side of the loading plot. The increased concentration of acetic acid (21) is identified as a key factor contributing to the intensification of sour and rancid sensory characteristics in grey jujube, while the increase in bitter attributes appears to be closely linked to changes in total flavonoid content. The variations in the correlating factors associated with bitterness warrant further investigation. These findings elucidate the distinct influence of individual physicochemical and aroma-active compounds on the sensory profile of grey jujube, offering insights into the chemical drivers behind quality changes during storage and contributing to a more comprehensive understanding of flavor stability in postharvest management.

## 4. Conclusions

The findings of this study highlight the profound effects of prolonged storage on the quality and sensory attributes of grey jujube fruit. Over the two- to three-year storage period, significant changes were observed in both color and texture, indicating a gradual decline in freshness and quality. The decrease in lightness and pigmentation of the peel, alongside the progressive darkening of the flesh, suggests that enzymatic oxidation processes are at play, impacting the visual appeal of the fruit. Physicochemical analyses revealed a notable reduction in moisture content, which directly correlated with increased hardness and chewiness, underscoring the role of moisture loss in altering textural properties. The degradation of sugars, particularly reducing sugars, and the increase in total acidity over time signal metabolic changes during storage, including potential implications for flavor and overall nutritional quality. Moreover, the analysis of active ingredients revealed a concerning decline in bioactive compounds such as vitamin C, total flavonoids, and cyclic adenosine monophosphate (cAMP), with significant oxidative degradation impacting their concentrations. While total polyphenols remained stable, the reductions in other antioxidants may compromise the fruit’s health benefits. Sensory evaluation further illustrated the deteriorating flavor profile, with a marked decrease in desirable attributes such as sweetness and fresh aroma, alongside an increase in off-flavors. These changes reflect the complex interplay of volatile compounds during storage, where desirable esters and aldehydes diminished significantly, while pungent acids, notably acetic acid, increased, indicating spoilage processes. In terms of sensory properties, the flavor profile deteriorated, with a noticeable decrease in desirable characteristics like sweetness and fresh aroma, accompanied by an increase in off-flavors. This shift was attributed to the reduction of key aroma-active compounds such as aldehydes and esters, while the increase in acetic acid indicated spoilage processes.

The observed changes in physicochemical components during the storage of grey jujube suggest the potential for using these alterations as markers for quality control. The variations in key bioactive compounds, such as flavonoids, reducing sugars, and acids, alongside the sensory changes, offer valuable insights for developing models to distinguish between newly harvested and aged grey jujubes. Such models could address a significant challenge in the industry by enabling the identification of newly harvested and aged grey jujubes based on the quality of the product, which would help ensure product quality and prevent the sale of substandard produce. This focus will guide our next research phase. Moreover, these findings underline the importance of optimizing storage conditions not only to preserve the fruit’s sensory appeal but also to maintain its nutritional value and health benefits. By focusing on these chemical markers, the grey jujube industry can enhance quality control measures, improve consumer satisfaction, and ultimately provide more reliable and fresh products to the market.

## Figures and Tables

**Figure 1 foods-14-00050-f001:**
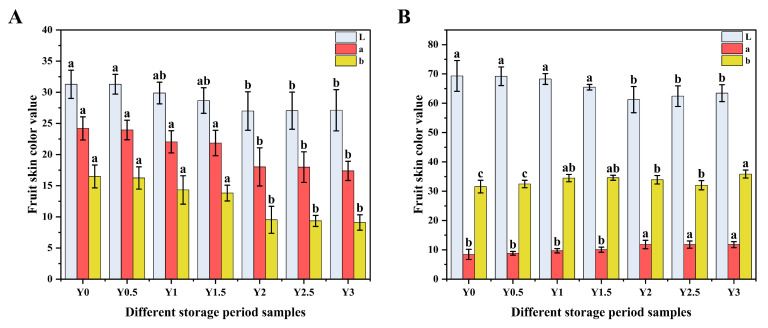
Color difference changes in grey jujube peel (**A**) and flesh (**B**) at different storage periods: (Y0) the pre-storage control sample; (Y0.5) samples stored for six months; (Y1) samples stored for one year; (Y1.5) samples stored for one and a half years; (Y2) samples stored for two years; (Y2.5) samples stored for two and a half years; (Y3) samples stored for three years; Different letters (a–c) in the figure indicate a significant difference between the two indicators at *p* < 0.05.

**Figure 2 foods-14-00050-f002:**
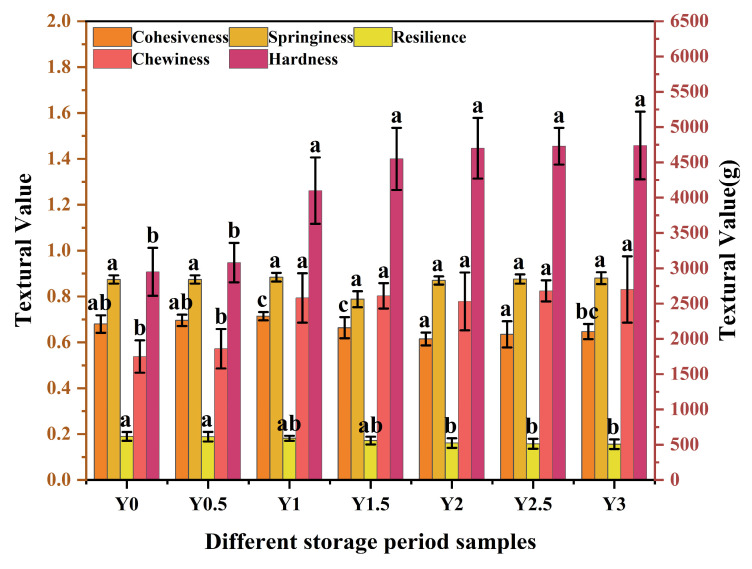
Textural characteristic changes of grey jujube at different storage periods. Hardness and chewiness are plotted on the right Y-axis, while springiness, cohesiveness, and resilience are shown on the left Y-axis: (Y0) the pre-storage control sample; (Y0.5) samples stored for six months; (Y1) samples stored for one year; (Y1.5) samples stored for one and a half years; (Y2) samples stored for two years; (Y2.5) samples stored for two and a half years; (Y3) samples stored for three years; Different letters (a–c) in the figure indicate a significant difference between the two indicators at *p* < 0.05.

**Figure 3 foods-14-00050-f003:**
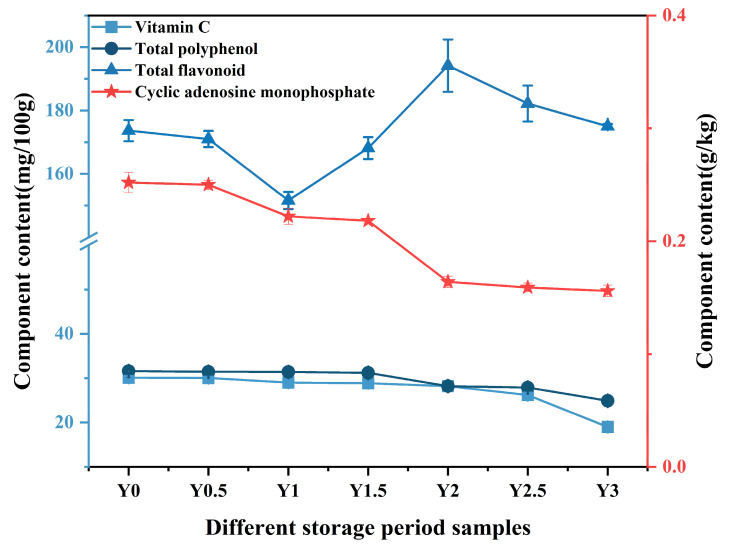
Antioxidant component changes of grey jujube at different storage periods. Vitamin C, Total polyphenols, and Total flavonoids are plotted on the left *Y*-axis, while Cyclic adenosine monophosphate is shown on the right *Y*-axis: (Y0) the pre-storage control sample; (Y0.5) samples stored for six months; (Y1) samples stored for one year; (Y1.5) samples stored for one and a half years; (Y2) samples stored for two years; (Y2.5) samples stored for two and a half years; (Y3) samples stored for three years.

**Figure 4 foods-14-00050-f004:**
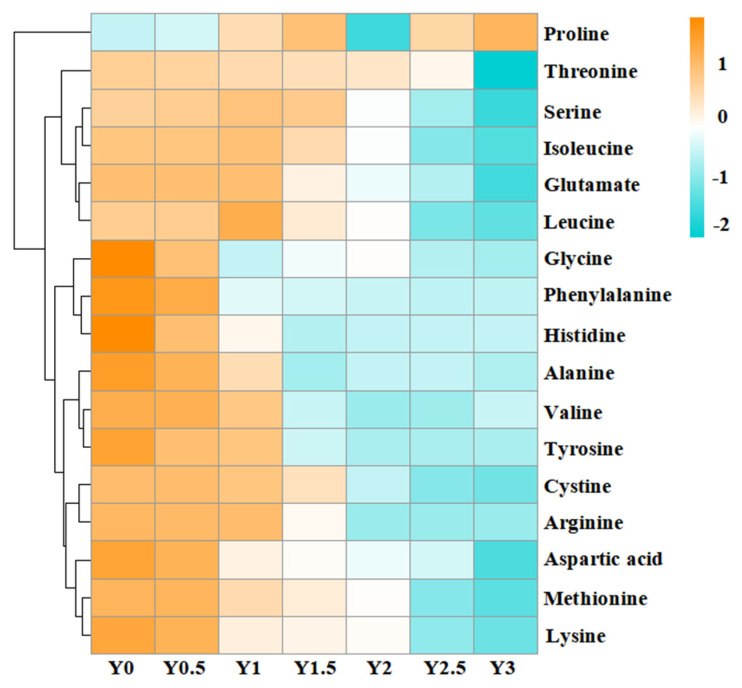
Cluster heatmap of amino acid content in grey jujube at different storage periods: (Y0) the pre-storage control sample; (Y0.5) samples stored for six months; (Y1) samples stored for one year; (Y1.5) samples stored for one and a half years; (Y2) samples stored for two years; (Y2.5) samples stored for two and a half years; (Y3) samples stored for three years.

**Figure 5 foods-14-00050-f005:**
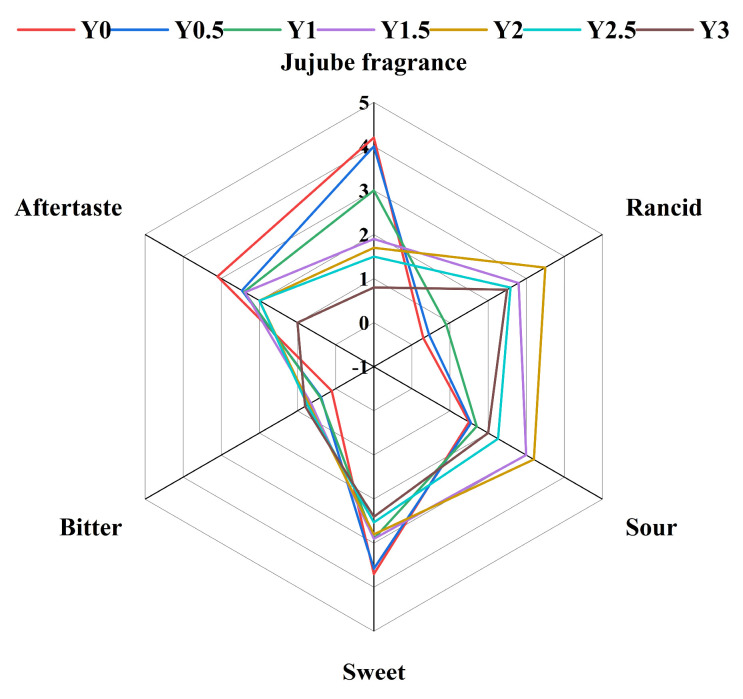
Spider chart of sensory evaluation for grey jujube at different storage periods: (Y0) the pre-storage control sample; (Y0.5) samples stored for six months; (Y1) samples stored for one year; (Y1.5) samples stored for one and a half years; (Y2) samples stored for two years; (Y2.5) samples stored for two and a half years; (Y3) samples stored for three years; The numbers on the vertical axis in the figure represent the sensory evaluation scores.

**Figure 6 foods-14-00050-f006:**
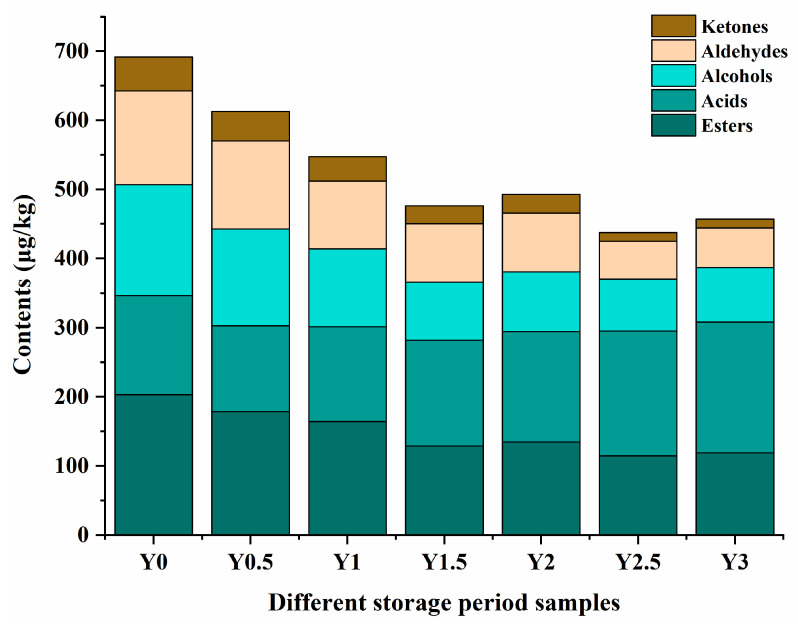
Effect of different storage times on flavor and composition content of grey jujube: (Y0) the pre-storage control sample; (Y0.5) samples stored for six months; (Y1) samples stored for one year; (Y1.5) samples stored for one and a half years; (Y2) samples stored for two years; (Y2.5) samples stored for two and a half years; (Y3) samples stored for three years.

**Figure 7 foods-14-00050-f007:**
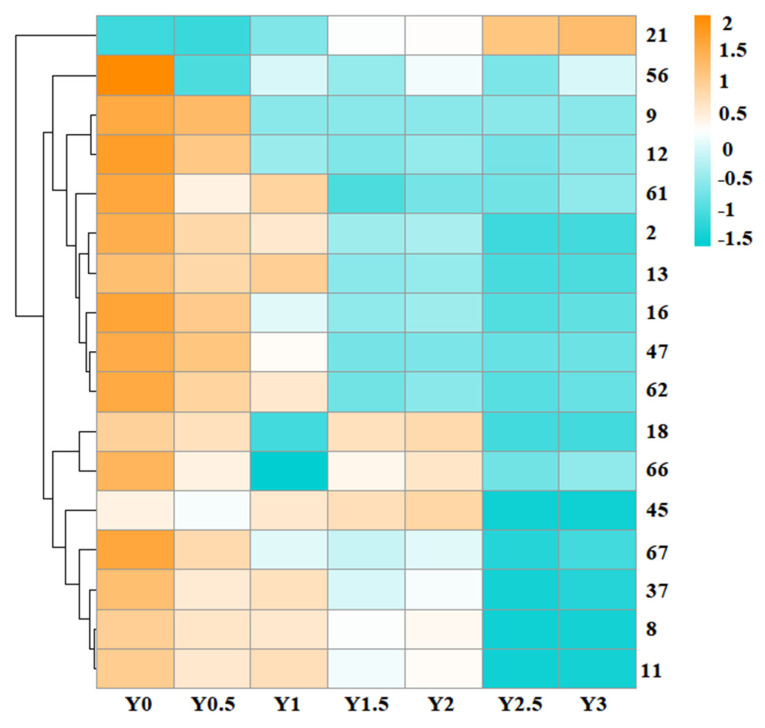
Cluster heatmap of key aroma component content in grey jujube at different storage periods: (Y0) the pre-storage control sample; (Y0.5) samples stored for six months; (Y1) samples stored for one year; (Y1.5) samples stored for one and a half years; (Y2) samples stored for two years; (Y2.5) samples stored for two and a half years; (Y3) samples stored for three years; The numbers on the right side of the heatmap correspond to the compounds listed with the same numbers in Table 2.

**Figure 8 foods-14-00050-f008:**
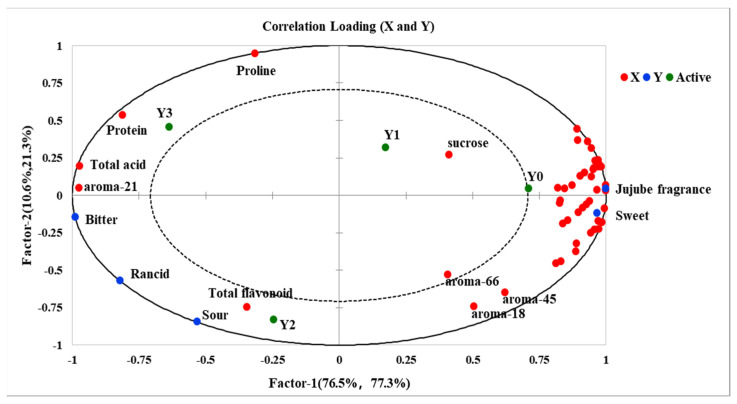
Correlation loadings for component content in grey jujube at different storage periods: (Y0) the pre-storage control sample; (Y0.5) samples stored for six months; (Y1) samples stored for one year; (Y1.5) samples stored for one and a half years; (Y2) samples stored for two years; (Y2.5) samples stored for two and a half years; (Y3) samples stored for three years; The inner and outer ellipses indicating 50% and 100% of the explained variance, respectively.

**Table 1 foods-14-00050-t001:** Chemical properties of grey jujube at different storage periods.

	Y0	Y0.5	Y1	Y1.5	Y2	Y2.5	Y3
Moisture content (g/100 g)	13.2 ± 0.4 a	13.2 ± 0.2 a	12.9 ± 0.3 b	12.5 ± 0.5 b	11.3 ± 0.3 c	11.0 ± 0.15 c	10.8 ± 0.3 c
Total sugar (g/100 g)	84.95 ± 0.68 a	83.68 ± 0.55 a	82.14 ± 0.50 b	82.11 ± 0.78 b	81.83 ± 0.32 b	81.50 ± 0.22 b	81.08 ± 0.85 b
Reducing sugar (g/100 g)	42.0 ± 0.4 a	41.97 ± 0.24 a	39.7 ± 0.5 b	39.1 ± 0.46 b	37.8 ± 0.6 c	36.42 ± 0.2 c	35.1 ± 0.3 d
Protein (g/100 g)	3.22 ± 0.20 b	3.35 ± 0.33 b	3.54 ± 0.13 ab	3.50 ± 0.24 ab	3.38 ± 0.08 ab	3.59 ± 0.15 ab	3.71 ± 0.14 a
Total acid (g/kg)	4.39 ± 0.27 d	4.50 ± 0.54 d	5.49 ± 0.16 c	5.88 ± 0.34 c	6.20 ± 0.07 b	6.94 ± 0.11 b	8.02 ± 0.04 a
Xylose (g/100g)	0.63 ± 0.01 a	0.60 ± 0.05 a	0.52 ± 0.03 b	0.49 ± 0.01 b	0.42 ± 0.02 c	0.38 ± 0.04 c	0.36 ± 0.02 c
Fructose (g/100 g)	17.26 ± 0.18 a	17.04 ± 0.14 a	16.87 ± 0.09 b	16.49 ± 0.05 b	16.54 ± 0.26 b	16.54 ± 0.18 b	16.54 ± 0.14 b
Glucose (g/100 g)	18.06 ± 0.05 a	17.92 ± 0.02 a	17.18 ± 0.03 b	17.08 ± 0.01 b	16.96 ± 0.14 b	16.96 ± 0.11 b	16.96 ± 0.07 b
Sucrose (g/100 g)	34.43 ± 0.13 a	34.18 ± 0.21 a	33.11 ± 0.14 b	33.09 ± 0.12 b	33.25 ± 0.20 b	33.24 ± 0.27 b	33.85 ± 0.18 b

a–d Values in the same column followed by different letters are significantly different at *p* < 0.05. Data are represented as the mean ± SD of at least a triplicate analysis; (Y0) the pre-storage control sample; (Y0.5) samples stored for six months; (Y1) samples stored for one year; (Y1.5) samples stored for one and a half years; (Y2) samples stored for two years; (Y2.5) samples stored for two and a half years; (Y3) samples stored for three years.

## Data Availability

The original contributions presented in the study are included in the article, further inquiries can be directed to the corresponding author.

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
