# Peer review of "Impact of Long-Term Cold Storage on the Physicochemical Properties, Volatile Composition, and Sensory Attributes of Dried Jujube (Zizyphus jujuba Mill.)"

_foods, 2024, doi:10.3390/foods14010050_

Round 1
Reviewer 1 Report
Comments and Suggestions for Authors
The topic deserves the attention.
Results are presented in a fair way.
English could be checked.
Latin name of Zizyphus jujuba Mill., should be written in Italic
All abbreviations should be written in full names when mentioned for the first time.
In Material and methods section, it would be nice to present photos of grey jujube fruits during storage period.
The main question is: Why did authors examined jujube samples stored for the period of three years.
Are jujube fruits sold on the market after being stored for three years?
This must be clarified in the Introduction.
My specific comments refer to the following:
Line 80-83: "... All fruits were selected after natural on-tree drying, characterized by a uniform red skin color. To ensure consistency across the samples, the jujubes were sun-dried under controlled conditions and manually sorted for uniform size and consistent color prior to experimentation."
Could you clarify this. Are fruits dried twice and why? Naturally and later sun-dried?
Line 83-85: The samples were stored in a controlled cold storage environment at 4°C for a duration of three years, maintaining stable conditions throughout the study.
What is cold storage environment? Be more precise.
What was the purpose of storing fruits for the period of three years?
Lines 85-90: "A total of seven sample groups were prepared based on different storage durations: the pre-storage control sample (Y0), samples stored for six months (Y0.5), one year (Y1), one and a half years (Y1.5), two years (Y2), two and a half years (Y2.5), and three years (Y3). Each sample group weighed approximately 3 kg. The weight of each sample was approximately 3 kg. All samples were packed in polyethylene bags, then stored at -20°C for analysis.
This is a complete mess. First authors mentioned that fruits were stored at 4 oC for three years, then at -20 oC for three years.
What is sample group? What is sample? Please clarify this!
Lines 129-131: "the results 129 were expressed as milligrams of gallic acid equivalents per 1,000 g of fresh weight (mg 130 GAE/1,000 g FW)"
Why did you express TPC on fresh weight when you stored dried samples?
Lines 466-468: Z and E isomers should be written in Italic. For example: (Z)-2-heptenal (11), … (E)-2-octenal (13),….
Line 496: Figure7. Cluster heatmap of key aroma component content in jujube at different storage periods
Numbers on the right side of the heatmap should be explained. What do they mean?
Line 517: The PLSR analysis indicates that the four samples (Y0, Y0.5, Y1, Y1.5, Y2, Y2.5, Y3)
It says four samples and in the bracket there are labels of 7 samples.
Please correct it.
Author Response
Reviewer Report 1:
The topic deserves the attention.
Results are presented in a fair way.
English could be checked.
Modify reply: Thank you very much for your recognition of our work. We fully acknowledge the questions you raised, and have made modifications and replied to all questions.The modified sections have been marked in red font in the document.
Latin name of Zizyphus jujuba Mill., should be written in Italic
Modify reply:Thank you for the reviewer's suggestions.The revision has been made as required.
All abbreviations should be written in full names when mentioned for the first time.
Modify reply: Thank you for the reviewer's suggestions.The revision has been made as required.The full forms of abbreviations such as GC-MS and GC-O have been provided at their first occurrence in the text.
In Material and methods section, it would be nice to present photos of grey jujube fruits during storage period.
Modify reply: Thank you for the reviewer's suggestions.Due to inconsistencies in the camera parameters during the photography process, the glossiness of the grey jujubes shown in the images differs slightly from the actual samples, which may lead to discrepancies in the analysis results. Therefore, the images were not included.
|
|
The main question is: Why did authors examined jujube samples stored for the period of three years.
Are jujube fruits sold on the market after being stored for three years?
This must be clarified in the Introduction.
Modify reply:Thank you for the reviewer's suggestions.We have provided an explanation in the Introduction. As our institution has been engaged in the reinspection of jujube futures for an extended period, distinguishing between newly harvested and stored jujubes has always been a challenge for the industry. This study investigates the quality changes of jujubes during storage from sensory and physicochemical perspectives, aiming to offer a new approach for differentiating between original and stored jujubes.
Revised Version:
Post-harvest, the majority of grey jujube production in Xinjiang is processed into dried fruit due to the challenges of fresh storage and transportation[7, 8]. Dried grey jujube is prized for its thick flesh, sweet and sour taste, and unique flavor, making it a favored product in the market[9, 10]. As the cultivation area and production volume of grey jujube in Xinjiang continue to expand, market pressures have intensified, leading to instances of unsold inventory during peak harvest seasons. To mitigate these challenges, an increasing number of enterprises have adopted cold storage during the harvest period, enabling off-season sales or utilizing the fruit for further processing. Currently, jujubes sold on the market can be stored for up to three years. Jujubes that have been stored for long periods are visually similar to original jujubes, making it difficult to distinguish their storage duration by appearance alone, which poses significant challenges to the development of the industry. However, despite its commercial success, storage presents significant challenges. Over time, the nutritional components of grey jujube degrade, leading to diminished sensory and nutritional quality, which in turn results in substantial economic losses[11-13]. While much of the research to date has focused on optimizing drying techniques and evaluating different jujube varieties, there is a relative paucity of studies examining the effects of long-term storage on the fruit’s nutritional and aromatic profiles.
My specific comments refer to the following:
Line 80-83: "... All fruits were selected after natural on-tree drying, characterized by a uniform red skin color. To ensure consistency across the samples, the jujubes were sun-dried under controlled conditions and manually sorted for uniform size and consistent color prior to experimentation."
Could you clarify this. Are fruits dried twice and why? Naturally and later sun-dried?
Modify reply: We sincerely thank the reviewer for raising this important question. There was an error in the original description, which has been corrected. Currently, grey jujubes circulating in the market are typically left to naturally dry on the tree for a period before harvesting. After being harvested, they are further dried by enterprises, primarily using hot air drying methods, until they reach a specific moisture content. The raw materials used in this study were procured from enterprises and had already undergone processing, making them marketable products.
Revised Version:
2.1. Materials
The grey jujube samples used in this study, specifically dried jujube (Zizyphus jujuba Mill.), were purchased from Xinjiang jujube industry Co., Ltd (Hetian, China), in 2019,with 13.2% water content and 84.95% total sugar content. All fruits were selected after natural on-tree drying, characterized by a uniform red skin color. To ensure consistency across the samples, the jujubes were dried under controlled conditions and manually sorted for uniform size and consistent color prior to experimentation. The samples were stored in in a cold storage facility at 4°C for three years, maintaining stable conditions throughout the study. During the storage period, samples were taken at regular intervals, resulting in a total of seven samples: the pre-storage control sample (Y0), samples stored for six months (Y0.5), one year (Y1), one and a half years (Y1.5), two years (Y2), two and a half years (Y2.5), and three years (Y3). The weight of each sample was approximately 3 kg. The samples taken from the cold storage were first used to select 40 fruits for measuring color and texture. 500 g of the sample was taken, chopped, and thoroughly mixed. Then, a subsample was taken using the quartering method, frozen with liquid nitrogen, and ground into a powder using a grinder. The powder was stored in a polyethylene plastic bottle at temperatures below -20°C for subsequent antioxidant measurements. The remaining samples were chopped, mixed thoroughly, and used for other physicochemical analyses.
Line 83-85: The samples were stored in a controlled cold storage environment at 4°C for a duration of three years, maintaining stable conditions throughout the study.
What is cold storage environment? Be more precise.
What was the purpose of storing fruits for the period of three years?
Modify reply: We sincerely thank the reviewer for raising this important question. Stored at 4°C under industry-standard conditions, the grey jujube were kept in cold storage for three years to investigate quality changes during the storage period, aiming to provide a theoretical basis for distinguishing between newly harvested and aged grey jujubes.
Lines 85-90: "A total of seven sample groups were prepared based on different storage durations: the pre-storage control sample (Y0), samples stored for six months (Y0.5), one year (Y1), one and a half years (Y1.5), two years (Y2), two and a half years (Y2.5), and three years (Y3). Each sample group weighed approximately 3 kg. The weight of each sample was approximately 3 kg. All samples were packed in polyethylene bags, then stored at -20°C for analysis.
This is a complete mess. First authors mentioned that fruits were stored at 4 oC for three years, then at -20 oC for three years.
What is sample group? What is sample? Please clarify this!
Modify reply: We sincerely apologize for any confusion caused. The samples were continuously stored in a cold storage facility at 4°C. At regular intervals, one sample (3 kg) was taken, resulting in a total of seven samples collected over the three-year storage period. Each collected sample was promptly transferred to a -20°C freezer for subsequent experiments. The manuscript has been revised accordingly.
Revised Version:
During the storage period, samples were taken at regular intervals, resulting in a total of seven samples: the pre-storage control sample (Y0), samples stored for six months (Y0.5), one year (Y1), one and a half years (Y1.5), two years (Y2), two and a half years (Y2.5), and three years (Y3). The weight of each sample was approximately 3 kg. All samples were packed in polyethylene bags and promptly stored in a -20°C freezer for subsequent experiments.
Lines 129-131: "the results 129 were expressed as milligrams of gallic acid equivalents per 1,000 g of fresh weight (mg 130 GAE/1,000 g FW)"
Why did you express TPC on fresh weight when you stored dried samples?
Modify reply:
Thank you for raising this question. The description in the manuscript was incorrect, and we sincerely apologize for the oversight. The total phenolic content (TPC) was actually expressed as milligrams of gallic acid equivalents per 1,000 grams of dry weight (mg GAE/1,000 g DW), not fresh weight as mistakenly stated. This has been corrected in the manuscript to accurately reflect the analysis conducted on the dried samples. We appreciate your careful review and understanding.
Lines 466-468: Z and E isomers should be written in Italic. For example: (Z)-2-heptenal (11), … (E)-2-octenal (13),….
Modify reply:Thank you for raising this question. We have thoroughly reviewed and revised the entire manuscript.
Line 496: Figure7. Cluster heatmap of key aroma component content in jujube at different storage periods
Numbers on the right side of the heatmap should be explained. What do they mean?
Modify reply:Thank you for raising this question.The numbers on the right side of the heatmap correspond to the compounds listed with the same numbers in Table 2.A note has been added to the figure legend.
Line 517: The PLSR analysis indicates that the four samples (Y0, Y0.5, Y1, Y1.5, Y2, Y2.5, Y3)
It says four samples and in the bracket there are labels of 7 samples.
Please correct it.
Modify reply: We sincerely appreciate the reviewer for identifying this issue, and the necessary revisions have been made in the manuscript.

Reviewer 2 Report
Comments and Suggestions for Authors
The work is valuable and interesting from a scientific point of view. However, several aspects need to be improved, and some sections need to be clarified. It is essential to improve my English and writing since several inconsistencies may be due to the translation of terms. Below I list my specific observations:
Abstract
Line 16: “renewed” - I don't know if the term in English is inappropriate, but it is understood to refer to the fact that the name was changed, so I think it may be a translation error. Please review.
Line 28: “The critical role of storage conditions”- It mentions the role of storage; however, the information provided above does not specify the storage conditions under which the study was conducted, which, in my opinion, would be essential to provide this information from the abstract.
Keywords
"Principal partial least square regression" is not a keyword; please review it and try to be more specific. Keywords should help you see the key concepts presented in the research, but this does not reflect that.
Introduction
Line 47-38: “medicinal properties and therapeutic potential”-Please specify which properties you are referring to.
Line 49: It refers to medicinal attributes again; the sentence is redundant; please correct the wording.
Line 53 and 55: Avoid expressions such as “higher and higher” and “more and more.” There are better ways to express your thoughts in more professional language.
Chemicals
Line 93-94: Does not correspond to chemicals.
Physical and Chemical composition analysis
Line 112: What is mentioned in this section corresponds only to chemical composition; there is no physical analysis.
Total phenolic content
Line 122: 5 g of sample-was the sample ground? Please specify how you did to homogenize it.
Line 125: change RPM to g
Line 130: Specify the range of concentrations at which the gallic acid standard curve was made.
Total flavonoid content
Line 136 and 137: Specify the concentration of NaOH you used to adjust the pH.
Vitamin C analysis
Line 152: 10-40 g of the slurry-10 g difference is a lot to set as a range. Was there no way to better control this part? Explain how the variation in the amount of slurry was countered.
Analysis of amino acids
Line 158: 2.000 g-Did you mean 2 g or is it 2,000 g? If it is 2 remove the zeros.
Line 160: Change rpm to g.
Cyclic adenosine monophosphate (cAMP) analysis
Line 164: “A 5 g sample”-5 g sample were…
Line 168: Change rpm to g
Line 172: 1.0 mL/min-1 mL/min.
Line 173: 50 mmol/L- 50 mM
Line 184: Mention xylose but not in the section title. Include it, please.
GC–MS analysis
Line 189: And what did this vial contain?
Line 197: 1.0 mL/min- 1 ml/min
Odor Activity Values (OAV)
Line 217: OA V-OAV, Is it with or without space? Please correct where appropriate.
Statistical procedures
Line 230-236: Only volatile constituent and aroma compounds were evaluated statistically?. Or why is it specified in this way? So, were not all the results obtained assessed in this way? Please clarify and complement if appropriate.
Results and Discussion
Physicochemical properties
Color analysis
Line 243: It mentions that there is a progressive darkening. To what can this behavior be attributed? In the end, he mentions something brief about it. Still, the discussion could be expanded by specifying the enzymes to which this activity is attributed, such as ethylene production, respiratory phenomena under storage conditions, etc.
Texture analysis
Figure 2. The left axis has no units
Nutritional properties
Generally, the sections in the results section are named as in the methodology; since the chemical composition is described in this section, the nutritional part would also have to be expressed in calories.
Antioxidant components
Figure 3. How do you explain the increase in flavonoids? It is one thing that they are protected and do not degrade, but how can you explain that they increase considerably after the first year of storage? Go Deeper
Changes of free amino acids
Line 340: “storage(Table2)”-storage (Table 2)
Line 357-371: Maillard reactions are mentioned repeatedly; however, this would not be possible at the storage temperature at which the samples were kept since this requires high temperatures. Review and modify the discussion regarding these parameters. Alternatively, explain in greater depth how it would be possible for this type of reaction to occur under these conditions.
Volatile aroma compounds
Profile of jujube flavor in different storage periods
Figure 6. esters is written in lowercase in the legend.
Changes of Aroma-Active Compounds during the jujube storage process
Line 460 and 466: Start the sentence with a capital letter
Principal partial least square regression (PLSR )
Line 502: “nine physicochemical parameters”-Please specify which ones.
Line 503: “four antioxidant-active compounds”-Please specify which ones. “Seventeen amino acids”-Please specify which ones. "seventeen pivotal flavor”-Please specify which ones.
Line 506—509: Provide the model equation.
Comments on the Quality of English LanguageEnglish should be greatly improved.
Author Response
Reviewer Report 2:
The work is valuable and interesting from a scientific point of view. However, several aspects need to be improved, and some sections need to be clarified. It is essential to improve my English and writing since several inconsistencies may be due to the translation of terms. Below I list my specific observations:
Modify reply: Thank you very much for your recognition of our work. We fully acknowledge the questions you raised, and have made modifications and replied to all questions.The modified sections have been marked in green font in the document.
Abstract
1、Line 16: “renewed” - I don't know if the term in English is inappropriate, but it is understood to refer to the fact that the name was changed, so I think it may be a translation error. Please review.
Modify reply: Thank you for the reviewer's suggestions. We have replaced the terminology in the manuscript.
Revised Version:
Grey jujube (Zizyphus jujuba Mill. cv. Huizao), a prominent cultivar from Xinjiang, China, is well-known for its high nutritional value and medicinal benefits.
2、Line 28: “The critical role of storage conditions”- It mentions the role of storage; however, the information provided above does not specify the storage conditions under which the study was conducted, which, in my opinion, would be essential to provide this information from the abstract.
Modify reply:Thank you for the reviewer's suggestions. The abstract indeed lacked a clear statement of the research purpose, so we have reorganized it to explicitly articulate the study's objectives. At present, the industry predominantly utilizes standardized conditions, notably 4°C cold storage, for the preservation of grey jujubes. This study investigates the quality changes of grey jujubes during storage from sensory and physicochemical perspectives, aiming to provide a novel approach for differentiating between newly harvested and aged grey jujubes. Furthermore, these findings offer theoretical support for improving and optimizing storage conditions.
Revised Version:
Abstract: Grey jujube (Zizyphus jujuba Mill. cv. Huizao), a prominent cultivar from Xinjiang, China, is well-known for its high nutritional value and medicinal benefits. This study investigates the effects of long-term storage on the quality attributes of grey jujube fruit, focusing on color, texture, physicochemical properties, bioactive ingredients, amino acid profiles, sensory characteristics, and volatile compounds. Over a three-year storage period, significant changes were observed, including a decline in lightness and redness of the peel, accompanied by textural modifications such as increased hardness and chewiness, primarily attributed to moisture loss. Physicochemical analyses revealed significant reductions in moisture content, sugars (particularly reducing sugars), and bioactive compounds such as vitamin C, total flavonoids, and cyclic adenosine monophosphate (cAMP). In contrast, total acidity exhibited a marked increase over time. Sensory evaluation indicated a substantial loss of fresh aroma and flavor, with the emergence of off-flavors over time. Additionally, a comprehensive analysis of volatile compounds highlighted significant transformations in aroma profiles, with notable reductions in desirable esters and increases in acetic acid concentrations. This study investigates the quality changes of grey jujubes during storage from sensory and physicochemical perspectives, aiming to provide a novel approach for differentiating between newly harvested and aged grey jujubes. Furthermore, these findings offer theoretical support for improving and optimizing storage conditions. Future research should focus on elucidating the underlying mechanisms of these changes, identifying key markers for quality control during grey jujube storage, and providing a scientific basis for distinguishing between newly harvested and aged grey jujubes, while improving storage quality.
3、Keywords
"Principal partial least square regression" is not a keyword; please review it and try to be more specific. Keywords should help you see the key concepts presented in the research, but this does not reflect that.
Modify reply:Thank you for the reviewer's suggestions. We have adjusted the keywords.
Revised Version:
Keywords: Grey jujube; Storage; Quality changes; Descriptive sensory analysis; Physicochemical-Sensory Correlation
Introduction
4、Line 47-38: “medicinal properties and therapeutic potential”-Please specify which properties you are referring to.
Modify reply:Thank you for the reviewer's suggestions.We have made the revisions as requested.
Revised Version:
Additionally, it possesses a distinct flavor profile and is abundant in essential trace elements and bioactive compounds, which contribute to its pharmacological effects, including sedative properties, hematopoietic support, antioxidant activity, and antineoplastic potential
5、Line 49: It refers to medicinal attributes again; the sentence is redundant; please correct the wording.
Modify reply:Thank you for the reviewer's suggestions.We have made the revisions as requested.
6、Line 53 and 55: Avoid expressions such as “higher and higher” and “more and more.” There are better ways to express your thoughts in more professional language.
Modify reply:Thank you for the reviewer's suggestions.We have made the revisions as requested.
Revised Version:
With the improvement of people's living standards, there is an increasing demand for higher fruit quality. As one of the key indicators of fruit quality, the nutritional and aroma components of fruit have garnered growing attention from researchers.
Chemicals
7、Line 93-94: Does not correspond to chemicals.
Physical and Chemical composition analysis
Modify reply:Thank you for the reviewer's suggestions.We have made the revisions as requested.
Revised Version:
2.2. Chemicals
A homologous series of n-alkanes (C8–C20) was utilized for the calculation of retention indices (RI) in the gas chromatography analyses. The experimental procedures, including RI calculations, were conducted following the manufacturer’s guidelines (Sigma Chemical Co., St. Louis, MO, USA). 2-Octanol, used as an internal standard at a concentration of 20 mg/kg, was obtained from J&K Chemical Ltd (Shanghai, China). Concentrated hydrochloric acid, sodium hydroxide, methyl red indicator, Fehling's reagent, potassium sodium tartrate, glacial acetic acid, zinc acetate, potassium ferrocyanide, sodium tungstate, sodium molybdate, sodium carbonate, copper sulfate, potassium sulfate, boric acid, concentrated sulfuric acid, bromocresol green indicator, methylene blue indicator, 95% ethanol, phenolphthalein indicator, metaphosphoric acid, sodium bicarbonate, and 2,6-dichlorindole, all of analytical grade, were obtained from Sinopharm Chemical Reagent Co. (Shanghai, China). Adenosine cyclic monophosphate, rutin, tannin, vitamin C, fructose, glucose, sucrose, xylose, threonine, serine, glutamate, glycine, alanine, cystine, valine, methionine, isoleucine, leucine, tyrosine, phenylalanine, lysine, histidine, arginine, and proline were purchased from Sigma-Aldrich (St. Louis, MO, USA).
8、Line 112: What is mentioned in this section corresponds only to chemical composition; there is no physical analysis.
Modify reply:Thank you for the reviewer's suggestions.We have made the revisions as required and removed "physical analysis."
Total phenolic content
- Line 122: 5 g of sample-was the sample ground? Please specify how you did to homogenize it.
Modify reply:Thank you for the reviewer's suggestions. In the "2.1. Materials" section, a description of the sample pretreatment has been added.
Revised Version:
2.1. Materials
The grey jujube samples used in this study, specifically dried jujube (Zizyphus jujuba Mill.), were purchased from Xinjiang jujube industry Co., Ltd (Hetian, China), in 2019,with 13.2% water content and 84.95% total sugar content. All fruits were selected after natural on-tree drying, characterized by a uniform red skin color. To ensure consistency across the samples, the jujubes were dried under controlled conditions and manually sorted for uniform size and consistent color prior to experimentation. The samples were stored in in a cold storage facility at 4°C for three years, maintaining stable conditions throughout the study. During the storage period, samples were taken at regular intervals, resulting in a total of seven samples: the pre-storage control sample (Y0), samples stored for six months (Y0.5), one year (Y1), one and a half years (Y1.5), two years (Y2), two and a half years (Y2.5), and three years (Y3). The weight of each sample was approximately 3 kg. The samples taken from the cold storage were first used to select 40 fruits for measuring color and texture. 500 g of the sample was taken, chopped, and thoroughly mixed. Then, a subsample was taken using the quartering method, frozen with liquid nitrogen, and ground into a powder using a grinder. The powder was stored in a polyethylene plastic bottle at temperatures below -20°C for subsequent antioxidant measurements. The remaining samples were chopped, mixed thoroughly, and used for other physicochemical analyses.
- Line 125: change RPM to g
Modify reply:Thank you for the reviewer's suggestions.We have made the revisions as required.
- Line 130: Specify the range of concentrations at which the gallic acid standard curve was made.
Modify reply:Thank you for the reviewer's suggestions.We have made the revisions as required.
Revised Version:
2.6 Total phenolic content
The total phenolic content of the grey jujube extracts was determined using the Folin–Ciocalteu method [1]. 5 g of sample was placed in a 100 mL volumetric flask and diluted with ultrapure water to 80 mL. The flask was immersed in boiling water for 30 minutes. After cooling to room temperature, the solution was diluted to the final volume. The extracts were then centrifuged (Sorvall ST8, Thermo Fisher Scientific, USA) at 4690 g for 4 minutes. A 1 mL aliquot of the supernatant was mixed with 5 mL of deionized water, 3 mL of 7.5% sodium carbonate (Na₂CO₃) solution, and 1 mL of Folin–Ciocalteu reagent. The mixture was thoroughly vortexed and incubated in the dark at room temperature for 2 hours. Absorbance was measured at 765 nm using a spectrophotometer, and the results were expressed as milligrams of gallic acid equivalents per 1,000 g of fresh weight (mg GAE/1,000 g DW). The concentration range for the standard curve of gallic acid is 0.5-5.0 mg/L.
Total flavonoid content
12、Line 136 and 137: Specify the concentration of NaOH you used to adjust the pH.
Modify reply:Thank you for the reviewer's suggestions.We have made the revisions as required.
Revised Version:
A 5 g sample was transferred to a 100 mL beaker, and the pH of the solution was adjusted to 13.0 by titration with sodium hydroxide (0.1mol/L NaOH).
Vitamin C analysis
- Line 152: 10-40 g of the slurry-10 g difference is a lot to set as a range. Was there no way to better control this part? Explain how the variation in the amount of slurry was countered.
Modify reply:Thank you for the reviewer's suggestions. This section was not written in a standardized manner. The 10-40 g range applies to different fruit and vegetable samples. Since grey jujube contains a relatively high level of vitamin C, a 10 g sample is typically sufficient. The text has been revised accordingly.
Analysis of amino acids
13、Line 158: 2.000 g-Did you mean 2 g or is it 2,000 g? If it is 2 remove the zeros.
Modify reply:Thank you for the reviewer's suggestions.The sample weight is 2 g, and the text has been revised as required.
14、Line 160: Change rpm to g.
Modify reply:Thank you for the reviewer's suggestions.We have made the revisions as required.
Cyclic adenosine monophosphate (cAMP) analysis
15、Line 164: “A 5 g sample”-5 g sample were…
Modify reply:Thank you for the reviewer's suggestions.We have made the revisions as required.
- Line 168: Change rpm to g
Modify reply:Thank you for the reviewer's suggestions.We have made the revisions as required.
17、Line 172: 1.0 mL/min-1 mL/min.
Modify reply:Thank you for the reviewer's suggestions.We have made the revisions as required.
18、Line 173: 50 mmol/L- 50 mM
Modify reply:Thank you for the reviewer's suggestions.We have made the revisions as required.
19、Line 184: Mention xylose but not in the section title. Include it, please.
Modify reply:Thank you for the reviewer's suggestions.We have made the revisions as required.
GC–MS analysis
20、Line 189: And what did this vial contain?
Modify reply:Thank you for the reviewer's suggestions.We have made the revisions as required.
Revised Version:
2.12 Gas chromatography-mass spectrometry (GC–MS) analysis
For each sample, 5 g of grey jujube puree was added to a 20-mL headspace bottle (Supelco, Bellefonte, PA, USA) supplemented with 20 μL 2-octanol of (32.88 μg/mL in methanol; Sigma-Aldrich, St Louis, MO) as an internal standard. The sample vial was closely capped with a PTFE-silicon stopper and equilibrated at 40 °C for 10 min. Then, a DVB/CAR/PDMS fiber was inserted into the headspace with continuous heating and agitation (250 rpm) for 30 min. The SPME extract of the Y0, Y1, Y2 and Y3 samples were injected into the port of an Thermos Trace1300-ISQ equipped with a DB-WAX capillary column (30 mm × 0.25 mm × 0.25 μm) and desorbed at 250°C for 5 min.
21、Line 197: 1.0 mL/min- 1 ml/min
Modify reply:Thank you for the reviewer's suggestions.We have made the revisions as required.
Odor Activity Values (OAV)
22、Line 217: OA V-OAV, Is it with or without space? Please correct where appropriate.
Modify reply:Thank you for the reviewer's suggestions.This is a writing error, which has been corrected.
Statistical procedures
23、Line 230-236: Only volatile constituent and aroma compounds were evaluated statistically?. Or why is it specified in this way? So, were not all the results obtained assessed in this way? Please clarify and complement if appropriate.
Modify reply: Thank you to the reviewer for pointing out this issue.There was an error in the description. All indicators were evaluated statistically.
Results and Discussion
Physicochemical properties
Color analysis
- Line 243: It mentions that there is a progressive darkening. To what can this behavior be attributed? In the end, he mentions something brief about it. Still, the discussion could be expanded by specifying the enzymes to which this activity is attributed, such as ethylene production, respiratory phenomena under storage conditions, etc.
Modify reply:Thank you for the reviewer's suggestions.We have added content to the discussion section, providing a more in-depth exploration of the mechanisms behind the darkening of grey jujube during storage.
Revised Version:
3.1.1 Color analysis
Color changes in the peel and flesh of grey jujube during storage were evaluated and are presented in Figure 1. Compared to the initial storage sample (Y0), significant decreases in L* (lightness), a* (redness), and b* (yellowness) values of the peel were observed (p < 0.05) after two years of storage(Y2), indicating a progressive darkening and fading of red hues. Specifically, the peel L* value decreased from 31.29 ± 2.26 (Y0) to 26.99 ± 3.10 (Y2) and 27.11 ± 3.31 (Y3), while the a* value dropped from 24.20 ± 1.86 (Y0) to 17.38 ± 1.54 (Y3), and the b* value declined from 16.49 ± 1.84 (Y0) to 9.10 ± 1.24 (Y3). These changes suggest that the peel underwent noticeable darkening and loss of red pigmentation over time.
For the grey jujube flesh, a gradual decrease in L* was also observed, with values dropping from 69.34 ± 5.24 (Y0) to 61.24 ± 4.46 (Y2), reflecting reduced lightness. Conversely, a* and b* values of the flesh increased over time, with the a* value rising from 8.42 ± 1.74 (Y0) to 11.76 ± 1.03 (Y3), and the b* value increasing from 31.55 ± 2.15 (Y0) to 35.83 ± 1.37 (Y3). These changes suggest a deepening of the flesh color, transitioning towards more intense red and yellow hues as storage progressed.
The observed color changes in both the peel and flesh of grey jujube during storage may be attributed to several factors. Firstly, grey jujube contains oxidase enzymes, such as polyphenol oxidase (PPO) and catalase (CAT), which can be activated during storage. These enzymes catalyze the oxidation of phenolic compounds, leading to changes in pigment composition and a subsequent darkening of the fruit. For instance, naturally occurring flavonoids and anthocyanins in jujube may undergo oxidation, contributing to the color alterations observed. This enzymatic oxidation process is consistent with findings reported in previous studies, where similar changes in color were attributed to the oxidative degradation of phenolic compounds. Additionally, the respiratory activity of jujubes during storage, which persists at a certain metabolic rate, consumes sugars and produces carbon dioxide and ethylene. These metabolic processes can further induce chemical changes within the fruit, influencing the color stability of both the peel and flesh. Consequently, this oxidative degradation, combined with metabolic activity, likely contributes to the progressive darkening of the peel and the intensification of the flesh coloration, highlighting the impact of long-term storage on the appearance and quality of grey jujube [28, 29].
Texture analysis
25、Figure 2. The left axis has no units
Modify reply: Since texture parameters such as springiness and cohesiveness are calculated through a series of area ratios and other methods, they are dimensionless and therefore do not have units.
Nutritional properties
26、Generally, the sections in the results section are named as in the methodology; since the chemical composition is described in this section, the nutritional part would also have to be expressed in calories.
Modify reply: Thank you to the reviewer for pointing out this issue.This has been revised to "Chemical properties" to align with the methodology section.
Antioxidant components
27、Figure 3. How do you explain the increase in flavonoids? It is one thing that they are protected and do not degrade, but how can you explain that they increase considerably after the first year of storage? Go Deeper
Modify reply:Thank you to the reviewer for pointing out this issue.
The increase in total flavonoid content during storage may be attributed to the loss of substances such as sugars, leading to a decrease in the total dry matter content. The specific mechanism requires further research and exploration.A preliminary discussion on this matter is provided in the manuscript.
Revised Version:
3.1.4 Antioxidant components
Four bioactive compounds were monitored throughout storage: vitamin C, total polyphenols, total flavonoids, and cyclic adenosine monophosphate (cAMP), as shown in Figure 3. Results indicate a significant decline in vitamin C, total polyphenols, and cAMP content over the three-year storage period, while total flavonoids levels showed no significant change. Specifically, vitamin C, total polyphenols, and cAMP decreased by 36.9%, 21.2%, and 38.1%, respectively, suggesting susceptibility to oxidative degradation under prolonged storage conditions. The reduction in vitamin C and polyphenols content may result from oxidative reactions triggered by residual enzymatic activity and ambient oxygen, processes which can concurrently impact color stability, leading to a progressive darkening of the fruit. These oxidative losses not only reduce the antioxidant capacity of grey jujube but also affect its sensory qualities and nutritional profile, potentially diminishing the fruit's perceived health benefits[29].
Interestingly, the stability of total flavonoids across the storage period contrasts with the decline observed in other antioxidants, suggesting that flavonoids in grey jujube may be more resistant to oxidation under cold storage. The increase in total flavonoid content after the first year of storage may be attributed to the loss of substances such as sugars, leading to a decrease in the total dry matter content. Flavonoid stability could be advantageous in maintaining some level of antioxidant protection. The stability of flavonoids during storage, particularly their resistance to oxidative degradation, could be attributed to several factors. Flavonoids possess a characteristic C6-C3-C6 structure, consisting of two aromatic rings (A and B) connected by a three-carbon bridge (C). This arrangement allows flavonoids to stabilize free radicals through their conjugated double bonds, enhancing their antioxidant properties. The presence of hydroxyl groups on the aromatic rings further contributes to their ability to neutralize oxidative agents. Additionally, glycosylation can affect their stability and bioavailability. The structural stability of flavonoids, particularly their aromatic ring system and hydroxyl groups, makes them resistant to oxidative degradation, thus preserving their antioxidant activity during cold storage. However, the precise mechanisms behind this stability, including potential protective interactions with other compounds in jujube, require further investigation. We plan to explore this aspect in more detail in future studies to better understand how specific flavonoid compounds contribute to maintaining antioxidant protection during storage.
Changes of free amino acids
- Line 340: “storage(Table2)”-storage (Table 2)
Modify reply:Thank you for the reviewer's suggestions.We have made the revisions as required.
29、Line 357-371: Maillard reactions are mentioned repeatedly; however, this would not be possible at the storage temperature at which the samples were kept since this requires high temperatures. Review and modify the discussion regarding these parameters. Alternatively, explain in greater depth how it would be possible for this type of reaction to occur under these conditions.
Modify reply:Thank you for the reviewer's suggestions.We have made the revisions as required.
Revised Version:
3.1.4 Changes of free amino acids
A total of 17 amino acids were quantified in grey jujube samples during storage(Table 2), with most showing a gradual decline over time, except for proline, which remained stable. The trend in amino acid content changes during storage was clearly visible in the clustering heatmap(Figure 4). Proline was the most abundant amino acid, accounting for nearly half of the total amino acid content. Significant changes were observed in aspartic acid, glycine, alanine, phenylalanine, lysine, and histidine, which showed notable reductions, whereas threonine and methionine levels remained relatively stable over the two-year storage period. Other amino acids, including serine, glutamate, cystine, valine, isoleucine, leucine, tyrosine, and arginine, exhibited no significant change in the first year but declined significantly after two years of storage.
The stability of proline can be attributed to two primary factors. First, proline’s unique secondary amine structure makes it less prone to undergo non-enzymatic browning reactions, under the ambient temperatures typically employed during storage. This structural resilience likely minimizes proline's involvement in such reactions, preserving its content over time. Second, although protein hydrolysis during storage could potentially generate proline, it seems that any contributions from this process are offset by concurrent degradation mechanisms, resulting in a relatively stable proline level throughout the storage period[13].
The observed reductions in other amino acids likely result from several storage-related factors. Amino acids with reactive side chains, such as lysine and arginine, are particularly prone to degradation through oxidative pathways, even at lower temperatures [32]. This degradation contributes not only to a reduction in amino acid content but also to the formation of compounds that may influence the fruit’s flavor, color, and nutritional profile. The gradual reduction in these amino acids may, therefore, impact both the organoleptic properties and the nutritional value of stored grey jujube, underscoring the potential quality decline over extended storage periods.
In summary, the stability of proline coupled with the degradation of other amino acids highlights the complex interplay of chemical reactions during storage, with implications for both the sensory quality and nutritional composition of grey jujube. These findings suggest that optimizing storage conditions to limit oxidative reaction activity could help preserve amino acid integrity, particularly for amino acids vulnerable to degradation. Further research focusing on storage interventions that minimize amino acid loss, could enhance the overall quality and nutritional retention in long-term stored grey jujube products.
Volatile aroma compounds
Profile of jujube flavor in different storage periods
30、Figure 6. esters is written in lowercase in the legend.
Modify reply:Thank you for the reviewer's suggestions.We have made the revisions as required.
Changes of Aroma-Active Compounds during the jujube storage process
31、Line 460 and 466: Start the sentence with a capital letter
Modify reply:Thank you for the reviewer's suggestions.We have made the revisions as required.
Principal partial least square regression (PLSR )
- Line 502: “nine physicochemical parameters”-Please specify which ones.
Modify reply:Thank you for the reviewer's suggestions.We have made the revisions as required.
- Line 503: “four antioxidant-active compounds”-Please specify which ones. “Seventeen amino acids”-Please specify which ones. "seventeen pivotal flavor”-Please specify which ones.
Modify reply:Thank you for the reviewer's suggestions.We have made the revisions as required.
Revised Version:
3.4 Principal partial least square regression (PLSR )
To elucidate the correlation between the physicochemical constituents, key aroma-active compounds, and sensory attributes of grey jujube, partial least squares regression (PLSR) analysis was conducted. The compounds selected for analysis were rigorously chosen based on their relevance to the study’s objectives, ensuring they represent critical influences on sensory qualities. Specifically, nine conventional physicochemical parameters(moisture content, total sugar, reducing sugar, protein , total acid, Xylose, fructose, glucose, sucrose ), four antioxidant-active components( vitamin C, total polyphenols, total flavonoids, and cyclic adenosine monophosphate), seventeen amino acids(threonine, serine, glutamate, glycine, alanine, cystine, valine, methionine, isoleucine, leucine, tyrosine, phenylalanine, lysine, histidine, arginine, and proline ), and seventeen key aroma compounds(listed in Figure 7) were incorporated as predictor variables in the PLSR model. Here, the X-matrix encompassed the chemical composition of the grey jujube samples, while the Y-matrix corresponded to the sensory attributes of interest. The developed PLS2 model comprised three principal components, accounting for 100 % of the cross-validated variance, indicating the model's effectiveness in accurately capturing the comprehensive information of the samples. The first two principal components (PC1 and PC2) were sufficient to represent the overall sample information, explaining 98.7% of the cross-validated variance, while additional components (PCs) contributed minimal useful information. Consequently, only the correlation loading plots for PC1 and PC2 are presented (Figure 8). As shown in Figure 8, nearly all variables fall within the two ellipses (R² = 0.5 and R² = 1.0), representing 50% and 100% of the explained variance, respectively, indicating that these variables are well-explained by the PLSR model. Furthermore, the 47 compounds significantly impacted one or more of the five sensory descriptors.
34、Line 506—509: Provide the model equation.
Modify reply:Thank you for the reviewer's suggestions.The purpose of the PLSR analysis is to explore the correlation between physicochemical properties and sensory qualities. The conclusions of the analysis are clearly illustrated in the correlation loadings plot. Since the model equation involves 57 variables and is relatively complex, would it be acceptable not to include it in this section?

Reviewer 3 Report
Comments and Suggestions for Authors
1. The paper presents a three-year study on the storage of jujube fruits, analyzing various parameters. While the findings are insightful, the study's impact could be strengthened by exploring the underlying mechanisms driving the observed changes (e.g., specific oxidative pathways) and offering more thorough comparisons with existing literature.
2. The title emphasizes "storage time" as the main variable but underrepresents the role of other key storage conditions (e.g., temperature, moisture control, and packaging). This may mislead readers about the study’s comprehensive scope. A more precise title should include references to "storage conditions" or specify the controlled cold storage aspect to better reflect the study's focus.
Suggested Title:
Impact of long-term cold storage on the physicochemical properties, volatile composition, and sensory attributes of dried jujube (Zizyphus jujuba mill.)
3. Abstract (line 18): change the word “active” for “bioactive”
4. Materials and Methods (line 81): Specify the sun-drying controlled conditions
5. Results (Table 1): Suggestion: Move the table of volatile compound concentrations to the supplementary file, as it is too large to include in the main text.
6. Were environmental factors, such as oxygen exposure and light, considered or controlled during storage to explain the observed quality changes?
7. Ensure consistent use of terms throughout the manuscript. For instance, consistently refer to "grey jujube" or "jujube" to avoid confusion.
8. Some references appear outdated or lack detail. Such as:
42. Available online: https://www.vcfonline.nl/VcfCompoundSearch.cfm (accessed on 7 September 2022). 693
43. Available online: http://www.odour.org.uk/index.html (accessed on 7 September 2022).
According to Foods guidelines, references should include specific details such as the webpage or resource's title, the author's or organization's name (if applicable), and the full access date.
9. Results (lines 324-329): Please clarify the mechanisms underlying the stability of flavonoids during storage, particularly their resistance to oxidative degradation.
10. Could the stability of certain amino acids and flavonoids indicate potential markers for quality control? If so, this could be highlighted as a key takeaway.
Author Response
Reviewer Report 3:
- The paper presents a three-year study on the storage of jujube fruits, analyzing various parameters. While the findings are insightful, the study's impact could be strengthened by exploring the underlying mechanisms driving the observed changes (e.g., specific oxidative pathways) and offering more thorough comparisons with existing literature.
Modify reply: Thank you for the reviewer's suggestions. The draft has incorporated varying degrees of mechanistic discussion across different sections, which have been highlighted accordingly, particularly in the parts concerning color, antioxidant components, and amino acids.
- The title emphasizes "storage time" as the main variable but underrepresents the role of other key storage conditions (e.g., temperature, moisture control, and packaging). This may mislead readers about the study’s comprehensive scope. A more precise title should include references to "storage conditions" or specify the controlled cold storage aspect to better reflect the study's focus.
Suggested Title:
Impact of long-term cold storage on the physicochemical properties, volatile composition, and sensory attributes of dried jujube (Zizyphus jujuba mill.)
Modify reply:Thank you for the reviewer's suggestions. We believe this is an excellent title and have adopted the one suggested by the reviewer. Once again, we sincerely thank the reviewer for their valuable input.
- Abstract (line 18):change the word “active” for “bioactive”
Modify reply:Thank you for the reviewer's suggestions, The revisions have been made in the manuscript.
- Materials and Methods (line 81):Specify the sun-drying controlled conditions
Modify reply:Thank you for the reviewer's suggestions. There was an error in the original description, which has been corrected. Currently, grey jujubes circulating in the market are typically left to naturally dry on the tree for a period before harvesting. After being harvested, they are further dried by enterprises, primarily using hot air drying methods, until they reach a specific moisture content. The raw materials used in this study were procured from enterprises and had already undergone processing, making them marketable products.
Revised Version:
2.1. Materials
The grey jujube samples used in this study, specifically dried jujube (Zizyphus jujuba Mill.), were purchased from Xinjiang jujube industry Co., Ltd (Hetian, China), in 2019,with 13.2% water content and 84.95% total sugar content. All fruits were selected after natural on-tree drying, characterized by a uniform red skin color. To ensure consistency across the samples, the jujubes were dried under controlled conditions and manually sorted for uniform size and consistent color prior to experimentation. The samples were stored in in a cold storage facility at 4°C for three years, maintaining stable conditions throughout the study. During the storage period, samples were taken at regular intervals, resulting in a total of seven samples: the pre-storage control sample (Y0), samples stored for six months (Y0.5), one year (Y1), one and a half years (Y1.5), two years (Y2), two and a half years (Y2.5), and three years (Y3). The weight of each sample was approximately 3 kg. All samples were packed in polyethylene bags and promptly stored in a -20°C freezer for subsequent experiments.
5.Results (Table 1): Suggestion: Move the table of volatile compound concentrations to the supplementary file, as it is too large to include in the main text.
Modify reply: Thank you for the reviewer's suggestions. Previous communication with the editor confirmed that the tables can be included in the main text. Next, we will continue to communicate with the editor to confirm the formatting requirements and determine the final placement of the tables.
- Were environmental factors, such as oxygen exposure and light, considered or controlled during storage to explain the observed quality changes?
Modify reply: We appreciate the reviewer for raising this important point. Currently, the storage conditions in our experiment are primarily based on the storage practices used in the industry for dried jujubes, with samples stored at 4°C. At this stage, the effect of light exposure has not been considered. Regarding the impact of modified atmosphere storage in cold storage, we have not addressed this aspect, as most industrial cold storage facilities do not typically have controlled atmosphere capabilities.
7.Ensure consistent use of terms throughout the manuscript. For instance, consistently refer to "grey jujube" or "jujube" to avoid confusion.
Modify reply: We appreciate the reviewer for raising this important point. We have made uniform revisions throughout the manuscript, adopting the term "grey jujube" for consistency.
- Some references appear outdated or lack detail. Such as:
- Available online: https://www.vcfonline.nl/VcfCompoundSearch.cfm (accessed on 7 September 2022). 693
- Available online: http://www.odour.org.uk/index.html (accessed on 7 September 2022).
According to Foods guidelines, references should include specific details such as the webpage or resource's title, the author's or organization's name (if applicable), and the full access date.
Modify reply: Thank you for the reviewer's suggestions.The manuscript has been updated with some references, incorporating the latest sources, and additional details have been provided for references 42 and 43.
Revised Version:
- VCF Online Database. Retrieved September 7, 2022, fromhttps://www.vcfonline.nl/VcfCompound cfm.
- The Odour Database. Retrieved September 7, 2022, from http://www.odour.org.uk/index.html.
- Results (lines 324-329):Please clarify the mechanisms underlying the stability of flavonoids during storage, particularly their resistance to oxidative degradation.
Modify reply: Thank you for the reviewer's suggestions. The manuscript now includes a discussion on the mechanisms underlying the stability of flavonoid compounds. The exact mechanisms behind this stability, including potential protective interactions with other compounds in jujube, require further investigation. We plan to explore this aspect in more detail in future studies.
Revised Version:
3.1.4 Antioxidant components
Four bioactive compounds were monitored throughout storage: vitamin C, total polyphenols, total flavonoids, and cyclic adenosine monophosphate (cAMP), as shown in Figure 3. Results indicate a significant decline in vitamin C, total flavonoids, and cAMP content over the three-year storage period, while total flavonoids levels showed no significant change. Specifically, vitamin C, total polyphenols, and cAMP decreased by 36.9%, 21.2%, and 38.1%, respectively, suggesting susceptibility to oxidative degradation under prolonged storage conditions. The reduction in vitamin C and polyphenols content may result from oxidative reactions triggered by residual enzymatic activity and ambient oxygen, processes which can concurrently impact color stability, leading to a progressive darkening of the fruit. These oxidative losses not only reduce the antioxidant capacity of grey jujube but also affect its sensory qualities and nutritional profile, potentially diminishing the fruit's perceived health benefits[29].
Interestingly, the stability of total flavonoids across the storage period contrasts with the decline observed in other antioxidants, suggesting that flavonoids in grey jujube may be more resistant to oxidation under cold storage. Flavonoid stability could be advantageous in maintaining some level of antioxidant protection. The stability of flavonoids during storage, particularly their resistance to oxidative degradation, could be attributed to several factors. Flavonoids possess a characteristic C6-C3-C6 structure, consisting of two aromatic rings (A and B) connected by a three-carbon bridge (C). This arrangement allows flavonoids to stabilize free radicals through their conjugated double bonds, enhancing their antioxidant properties. The presence of hydroxyl groups on the aromatic rings further contributes to their ability to neutralize oxidative agents. Additionally, glycosylation can affect their stability and bioavailability. The structural stability of flavonoids, particularly their aromatic ring system and hydroxyl groups, makes them resistant to oxidative degradation, thus preserving their antioxidant activity during cold storage. However, the precise mechanisms behind this stability, including potential protective interactions with other compounds in jujube, require further investigation. We plan to explore this aspect in more detail in future studies to better understand how specific flavonoid compounds contribute to maintaining antioxidant protection during storage.
- Could the stability of certain amino acids and flavonoids indicate potential markers for quality control? If so, this could be highlighted as a key takeaway.
Modify reply: Thank you for your insightful suggestion.The identification of potential markers for quality control and the development of a model for distinguishing between newly harvested and aged grey jujubes using these markers is our next research plan. In the conclusion section of the manuscript, we have added a discussion on the feasibility of using these physicochemical properties as potential markers for quality control.
- Conclusions
The findings of this study highlight the profound effects of prolonged storage on the quality and sensory attributes of grey jujube fruit. Over the two- to three-year storage period, significant changes were observed in both color and texture, indicating a gradual decline in freshness and quality. The decrease in lightness and pigmentation of the peel, alongside the progressive darkening of the flesh, suggests that enzymatic oxidation processes are at play, impacting the visual appeal of the fruit. Physicochemical analyses revealed a notable reduction in moisture content, which directly correlated with increased hardness and chewiness, underscoring the role of moisture loss in altering textural properties. The degradation of sugars, particularly reducing sugars, and the increase in total acidity over time signal metabolic changes during storage, including potential implications for flavor and overall nutritional quality. Moreover, the analysis of active ingredients revealed a concerning decline in bioactive compounds such as vitamin C, total flavonoids, and cyclic adenosine monophosphate (cAMP), with significant oxidative degradation impacting their concentrations. While total polyphenols remained stable, the reductions in other antioxidants may compromise the fruit's health benefits. Sensory evaluation further illustrated the deteriorating flavor profile, with a marked decrease in desirable attributes such as sweetness and fresh aroma, alongside an increase in off-flavors. These changes reflect the complex interplay of volatile compounds during storage, where desirable esters and aldehydes diminished significantly, while pungent acids, notably acetic acid, increased, indicating spoilage processes. In terms of sensory properties, the flavor profile deteriorated, with a noticeable decrease in desirable characteristics like sweetness and fresh aroma, accompanied by an increase in off-flavors. This shift was attributed to the reduction of key aroma-active compounds such as aldehydes and esters, while the increase in acetic acid indicated spoilage processes.
The observed changes in physicochemical components during the storage of grey jujube suggest the potential for using these alterations as markers for quality control. The variations in key bioactive compounds, such as flavonoids, reducing sugars, and acids, alongside the sensory changes, offer valuable insights for developing models to distinguish between newly harvested and aged grey jujubes. Such models could address a significant challenge in the industry by enabling the identification of newly harvested and aged grey jujubes based on the quality of the product, which would help ensure product quality and prevent the sale of substandard produce.This will be the focus of our next research phase. Moreover, these findings underline the importance of optimizing storage conditions not only to preserve the fruit’s sensory appeal but also to maintain its nutritional value and health benefits. By focusing on these chemical markers, the grey jujube industry can enhance quality control measures, improve consumer satisfaction, and ultimately provide more reliable and fresh products to the market.

Reviewer 4 Report
Comments and Suggestions for Authors
The authors propose a manuscript titled “Effects of storage time on physicochemical properties, volatile composition, and sensory profile of jujube (Zizyphus jujuba Mill. cv. Huizao) fruits”
I suggest the following changes:
Keywords: 2 out of 5 keywords are already in the title, please try to find some alternate keywords.
Introduction: The authors should provide more recent literature references.
Materials and Methods
Please provide a reference for the moisture and sugar content standards.
For vitamin C and amino acid analysis please provide:
a) units and the equations used for the calculations,
b) number of replications.
For the sensory evaluation of jujubes please describe the scale used for each parameter.
Results and discussion should be extended and enriched with more references in order to support the results.
Please provide ANOVA tables with the mean squares and the significance with ** or * for 99% and 95% significance, respectively
I suggest to add table 3 as a supplementary file, due to its size
Table and figure legends should be more detailed.
Comments on the Quality of English LanguageThe entire manuscript needs an extensive English review.
Author Response
Reviewer Report 4:
The authors propose a manuscript titled “Effects of storage time on physicochemical properties, volatile composition, and sensory profile of jujube (Zizyphus jujuba Mill. cv. Huizao) fruits”
I suggest the following changes:
Modify reply:Thank you very much for your recognition of our work. We fully acknowledge the questions you raised, and have made modifications and replied to all questions.The modified sections have been marked in purple font in the document.
Keywords: 2 out of 5 keywords are already in the title, please try to find some alternate keywords.
Modify reply: Thank you for the reviewer's suggestions. We have revised the keywords accordingly.
Revised Version:Keywords: Jujube; Storage; Quality changes; Descriptive sensory analysis; Principal partial least square regression (PLSR )
Introduction: The authors should provide more recent literature references.
Modify reply:Thank you for the reviewer's suggestions.We have reorganized references [6], [8], [9], [10] and updated them with the most recent literature.
Materials and Methods
Please provide a reference for the moisture and sugar content standards.
Modify reply: Thank you for the reviewer's suggestions. We have added descriptions of moisture and total sugar content in the raw material introduction.
Revised Version: The grey jujube samples used in this study, specifically dried jujube (Zizyphus jujuba Mill.), were sourced from an orchard in Alar, Xinjiang, in 2019,with 13.2% water content and 84.95% total sugar content..
For vitamin C and amino acid analysis please provide:
- a) units and the equations used for the calculations,
- b) number of replications.
Modify reply: The calculation formula has been added as required, and detailed descriptions of the units and repetitions have been provided.
Revised Version: 2.8 Vitamin C analysis
Vitamin C content was determined using the visual titration method based on the reduction of 2,6-dichlorophenol-indophenol dye [18]. A 100 g portion of the edible part of the sample was weighed and placed into a tissue grinder, followed by the addition of 100 mL of 2% oxalic acid extraction solution. The sample was quickly ground into a homogeneous slurry. A portion of 10-40 g of the slurry was transferred into a 100 mL volumetric flask, diluted to the mark with extraction solution, and mixed thoroughly. The solution was filtered, and 10 mL of the filtrate was transferred into a 50 mL conical flask. Titration was performed using a standardized 2,6-dichlorophenol-indophenol solution until the solution turned pink and maintained the color for 15 seconds without fading. All determinations were done in triplicate, and results were expressed as milligrams per 100 g dm.
|
(1) |
In the formula, X represents the content of Vitamin C in mg/100g, V represents the volume of dye solution consumed during the titration of the sample solution in mL, V 0 represents the volume of dye solution consumed during the titration of the blank in mL, T represents the titration value of 2,6-dichlorindole dye(0.0952 mg/mL), A represents dilution factor, m represents sample weight in g, 100 represents unit conversion factor.
2.9 Analysis of amino acids
Amino acids were extracted from a 2.000 g jujube sample using 60 mL of 0.266 mol/L sulfosalicylic acid solution at room temperature over a 12-hour period. The resulting suspension was centrifuged at 5,000 rpm for 10 minutes at 0°C, then filtered through a 0.22 μm syringe filter. Amino acid content was quantified using the ninhydrin method on an automatic amino acid analyzer (A300, Membrapure, Germany). All determinations were done in triplicate, and results were expressed as grams per 100 g dm.
|
(2) |
In the formula, Xi represents the concentrations ofamino acids in g/100g, ci represents the content of amino acid i in the sample solution in nmol/mL, F represents dilution factor, V represents final volume in mL, M represents molar mass of amino acid i in g/mol, m represents sample weight in g, 109 and 100 represents unit conversion factor.
For the sensory evaluation of jujubes please describe the scale used for each parameter.
Modify reply: Thank you for the reviewer's suggestions. The revisions have been made as required, and the term "the scale" has been added to the sensory descriptors.
Revised Version:
2.15 Sensory evaluation of jujubes
The sensory characteristics of the jujube samples were evaluated by a trained panel of 10 members (5 males and 5 females) with extensive experience in food sensory evaluation, all recruited from the fruit and vegetable processing team. Organoleptic descriptors were quantified using six sensory attributes: “Jujube fragrance,” “Rancid,” “Sour,” “Sweet,” “Bitter,” and “Aftertaste” to assess both aroma defects and positive qualities. Each jujube sample was assigned a unique three-digit code and presented in random order to minimize bias. The evaluation was conducted at room temperature, with individual assessments recorded for each panelist.
Experienced assessors employed a 5-point scale, ranging from 0 to 5 with 1-unit increments, to evaluate each sensory attribute. A 3-minute break was provided between the evaluation of each sample. This scale is commonly used by Chinese panelists due to its simplicity and clarity, and it effectively captures the full spectrum of sensory intensities. Additionally, the scale includes enough discrete points to highlight subtle differences in intensity between samples. The results from three independent QDA tests for each odor descriptor were averaged and presented in a spider web chart.
Results and discussion should be extended and enriched with more references in order to support the results.
Modify reply: Thank you for the reviewer's suggestions. The draft has incorporated varying degrees of mechanistic discussion across different sections, which have been highlighted accordingly, particularly in the parts concerning color, antioxidant components, and amino acids.
Please provide ANOVA tables with the mean squares and the significance with ** or * for 99% and 95% significance, respectively
Modify reply: Thank you for the reviewer's suggestions. In both Table 1 and Table 2, the data are presented as mean ± SD, and different letters are used to indicate significant differences at P < 0.05.
I suggest to add table 3 as a supplementary file, due to its size
Modify reply: Thank you for the reviewer's suggestions. Previous communication with the editor confirmed that the tables can be included in the main text. Next, we will continue to communicate with the editor to confirm the formatting requirements and determine the final placement of the tables.
Table and figure legends should be more detailed.
Modify reply: Thank you for the reviewer's suggestions.We have provided detailed descriptions of the sample numbers in the captions of the tables and figures.

Round 2
Reviewer 1 Report
Comments and Suggestions for Authors
Accept
Author Response
Comments 1:Accept.
Response 1: Thank you very much for your recognition of our work.
Reviewer 3 Report
Comments and Suggestions for Authors
The authors have addressed the suggested revisions, allowing publication by Foods.
Author Response
Comments1:The authors have addressed the suggested revisions, allowing publication by Foods.
Response 1: Thank you very much for your recognition of our work.
Reviewer 4 Report
Comments and Suggestions for Authors
Please provide ANOVA tables with the mean squares and the significance with ** or * for 99% and 95% significance, respectively
Comments on the Quality of English LanguageMinor editing of English language required
Author Response
Comments1:
Please provide ANOVA tables with the mean squares and the significance with ** or * for 99% and 95% significance, respectively
Response 1: Thank you for the reviewer’s insightful suggestions. In both Table 1 and Table 2, the data are presented as mean ± SD, and different letters are used to indicate significant differences at P < 0.05. This approach effectively conveys the statistical comparisons and highlights significant differences in a clear and concise manner. As such, we believe the current presentation already fulfills the objective of illustrating the results and aligns with common practices in our field.
We hope this explanation is satisfactory and appreciate your understanding.
Comments2: Minor editing of English language required
Response 2: Thank you for the reviewer’s insightful suggestions. We have reviewed and revised the language throughout the manuscript, with the modifications highlighted in orange.